# Prospect of acromegaly therapy: molecular mechanism of clinical drugs octreotide and paltusotine

Jie Zhao [1,6], Hong Fu[1,6], Jingjing Yu[1,6], Weiqi Hong [1,6], Xiaowen Tian [1,6], Jieyu Qi[2,6], Suyue Sun[1], Chang Zhao [1], Chao Wu[1], Zheng Xu[1], Lin Cheng[3], Renjie Chai[2,3,4] ✉, Wei Yan [1] ✉, Xiawei Wei [1] ✉ & Zhenhua Shao [1,5] ✉

Somatostatin receptor 2 (SSTR2) is highly expressed in neuroendocrine tumors and represents as a therapeutic target. Several peptide analogs mimicking the endogenous ligand somatostatin are available for clinical use, but poor therapeutic effects occur in a subset of patients, which may be correlated with subtype selectivity or cell surface expression. Here, we clarify the signal bias profiles of the first-generation peptide drug octreotide and a new-generation small molecule paltusotine by evaluating their pharmacological characteristics. We then perform cryo-electron microscopy analysis of SSTR2-Gi complexes to determine how the drugs activate SSTR2 in a selective manner. In this work, we decipher the mechanism of ligand recognition, subtype selectivity and signal bias property of SSTR2 sensing octreotide and paltusotine, which may aid in designing therapeutic drugs with specific pharmacological profiles against neuroendocrine tumors.

Neuroendocrine tumors (NETs) are rare and serious tumors that are commonly caused by the excessive secretion of hormones, which result in complicated clinical syndromes and disorders[1,2]. For example, pituitary adenoma is a tumor of the pituitary gland that produces too much growth hormone (GH) and can lead to acromegaly[3]. Treatment of NETs include strategies that inhibit excess hormones and block hormone action[4,5].

Somatostatin receptors (SSTRs) are members of the G protein-coupled receptor (GPCR) superfamily and contain five subtypes (SSTR1–5). SSTRs, especially SSTR2, are expressed at high levels in NETs and have emerged as important biomarkers[4,6]. SSTRs and their endogenous ligands, somatostatin (SST) peptides, exert important physiological regulation of neuroendocrine functions[7]. Previous

studies have shown that SSTs and analogs are linked to decreased GH secretion, suggesting that the SSTR family may be a major therapeutic target for NETs[1]. However, the use of SST for disease treatment is limited owing to its short half-life in the body. A series of long-acting SST analogs have been developed and explored in clinical trials[8]. The first-generation drug octreotide is a biologically stable synthesized SST analog used for the treatment of acromegaly[7,9]. However, ~30% of the acromegalic patients do not respond well to octreotide[10]. Second-generation drugs have been developed and were recently reported to exhibit high safety profile[11]. Notably, agents with subtype selectivity have always been pursued in drug discovery targeting SSTR2, which could facilitate to improve the efficacy of potential drugs and avoid of side effects caused by off-target[4,7,12]. After much effort has been made

[1]Division of Nephrology and Kidney Research Institute, Laboratory of Aging Research and Cancer Drug Target, State Key Laboratory of Biotherapy and Cancer Center, National Clinical Research Center for Geriatrics, West China Hospital, Sichuan University, Chengdu, Sichuan 610041, China. [2]State Key Laboratory of Bioelectronics, Department of Otolaryngology Head and Neck Surgery, Zhongda Hospital, School of Life Sciences and Technology, Advanced Institute for Life and Health, Jiangsu Province High-Tech Key Laboratory for Bio-Medical Research, Southeast University, Nanjing 210096, China. [3]Department of Otolaryngology Head and Neck Surgery, Sichuan Provincial People's Hospital, University of Electronic Science and Technology of China, Chengdu, China. [4]Co-Innovation Center of Neuroregeneration, Nantong University, Nantong 226001, China. [5]Department of Nephrology, Hainan General Hospital, Haikou, Hainan 570311, China. [6]These authors contributed equally: Jie Zhao, Hong Fu, Jingjing Yu, Weiqi Hong, Xiaowen Tian, Jieyu Qi. ✉e-mail: renjiec@seu.edu.cn; weiyan2018@scu.edu.cn; xiaweiwei@scu.edu.cn; zhenhuashao@scu.edu.cn

over years, an SSTR2 subtype-selective agonist paltusotine has recently been developed by Crinetics, which is an orally bioavailable non-peptide small molecule and is undergoing phase 3 trials for the clinical treatment of acromegaly and other NETs.

SSTR2 transmits extracellular signals into cells through $G\alpha_{i/o}$ coupling and β-arrestin recruitment pathways. cAMP inhibition causes decreased hormone secretion and cell proliferation[13,14], while the β-arrestin recruitment results in subsequent receptor internalization thus influence the cell membrane distribution level and desensitization of SSTR2, which may lead to drug resistance and poor clinical curative effect[15–18]. Previous clinical evidence showed that the expression level of SSTR2 occurred down-regulated during the administration of SST analog octreotide for acromegalic patients[19], and lower of β-arrestin 1 in pituitary adenomas was associated with better drug efficacy[20]. It appears that available agents represent diverse therapeutic windows, probably due to their different pharmacological profiles. Therefore, understanding of the pharmacological characteristics and molecular mechanism of different ligands, including peptide and small-molecule drugs, may help improve therapeutic strategies involving SSTR2 signaling. Recently, the structures of SSTR2 bound to multiple ligands have been reported, however the subtype selectivity of ligand for distinguishing group 2 SSTRs and the bias property remain elusive[21–25].

Here, we explore the pharmacological features and molecular mechanisms of different SST ligands, including peptide and small-molecule drugs, with SSTR2. We find that the small molecule paltusotine possesses a better Gi-biased property compared with octreotide. The structures of SSTR2-Gi signaling complexes bind to octreotide or paltusotine are determined by cryo-electron microscopy (cryo-EM). From these results combined with the mutagenesis and cell-based functional assays, we elucidate the mechanism of how different drugs activate SSTR2 signaling and uncover a minor pocket in the paltusotine-bound structure. Furthermore, the subtype selectivity against SSTRs and bias property towards SSTR2 of paltusotine are deciphered. Our findings provide insights into the molecular mechanism of the promising clinical small-molecule drug for the treatment of NETs.

## Results

### Pharmacological characterization of octreotide and paltusotine

Octreotide is a cyclic octapeptide with pharmacophore mimicking SST14 (cyclic tetradecapeptide somatostatin-14), and the new-generation paltusotine is a selective non-peptide agonist that is apparently smaller than octreotide in size (Fig. 1a). We explored the pharmacological features of octreotide and paltusotine using HEK293 cells expressing SSTR2. cAMP accumulation inhibition assays revealed that the drugs displayed a similar ability to activate G protein-dependent signaling via SSTR2 (Fig. 1b). In β-arrestin recruitment assays, paltusotine exhibited a lower efficacy of β-arrestin recruitment, resulting in less receptor internalization, compared with octreotide (Fig. 1c–f). These results demonstrate that paltusotine is a better G-protein-biased agonist of SSTR2 compared with octreotide. These findings are consistent with a previous report showing that paltusotine may have better clinical application than octreotide and suggest that the level of receptor internalization may be associated with the therapeutic efficacy of the drugs[16,26]. Therefore, understanding the mechanism of ligand recognition and activation would help determining the bias feature of SSTR2 in response to diverse ligands.

### Recognition mechanism of octreotide by SSTR2

We next assembled the SSTR2-Gi1 complex by co-expression of the SSTR2 receptor with human $G\alpha_{i1}$ and human $G\beta_1\gamma_2$ in Sf9 cells. The complex structures of octreotide-bound and paltusotine-bound SSTR2 coupled to the Gi1 heterotrimer were determined by cryo-EM with a resolution of 3.37 and 3.24 Å, respectively (Fig. 2a–d, Supplementary Figs. 1–2 and Supplementary Table 1). The high-quality density enabled us to build the structural models accurately, including key residues for ligand recognition (Supplementary Fig. 3). The overall structures of the two complexes were very similar, with a root mean square deviation value of 0.76 Å for the Cα within the receptor and 1.25 Å for the receptor complex. Compared with the inactive structure of SSTR2 (PDB code: 7UL5), the structure of agonists bound SSTR2 exhibit obvious rearrangements of the key switches and TM5/TM6 required for G-protein coupling (Supplementary Fig. 4a), suggesting that the SSTR2-Gi1 signaling complexes with agonists adopt an active conformation.

The well-defined density in the orthosteric binding pocket allowed us to model octreotide unambiguously. Octreotide is a cyclic peptide that is bridged by a disulfide bond formed by cysteine at position 2 and position 7 (Fig. 2e, f). The main region of octreotide occupies the orthosteric binding site comprised by TM2, TM3, TM5, TM6, and TM7; the terminal region of octreotide is located in the extracellular vestibule formed by ECL2 and ECL3 (Fig. 2e–i). The binding of SSTR2 with octreotide exhibits the same recognition mechanism compared with previous reporting[21,22] (Supplementary Fig. 4b, c). (D)-Phe1 of octreotide forms hydrophobic interactions with I284 and P286 in ECL3 of SSTR2. The disulfide bond forms van der Waals forces with the side chain of F294[7.35], while the main chain of Cys2 makes polar contacts with the side chains of N276[6.55] and S279[6.58]. Phe3 forms hydrophobic interactions with I195[45.52] and Y205[5.35] in the extracellular end of TM5 (Fig. 2g). Octreotide contains the same pharmacological core region ((D)-Tyr4 and Lys5) as that in SST14, the side chains of (D)-Tyr4 and Lys5 insert into the bottom of the orthosteric site of the receptor and are necessary for receptor activation (Supplementary Fig. 4d). Furthermore, (D)-Tyr4 is accommodated in a hydrophobic pocket formed by residues F127[3.37], F208[5.38] and F272[6.51], and Lys5 makes direct interactions with the side chains of D122[3.32], Q126[3.36], and Y302[7.43] via a polar network (Fig. 2i). Position 3.32 (D or E) was reported to be a key conserved residue involved in ligand binding and activation in aminergic receptors or peptide receptors[27,28]. Notably, (D)-Tyr4 and Lys5 in octreotide or SST analogs are the core region of the cyclic peptide. Alanine replacement of the residues F208[5.38], F272[6.51], Y302[7.43], Q126[3.36], and D122[3.32] in SSTR2 significantly decreased the cAMP inhibition potency (Fig. 2j, k, Supplementary Fig. 5 and Supplementary Table 2), suggesting that the interactions of (D)-Tyr4 and Lys5 from the core region with the receptor are essential for the biological activity of the hormone SST or octreotide.

Octreotide bound to the orthosteric binding pocket by extensive interactions with SSTR2; this ligand retains high binding affinity and efficacy with the SSTR family members, including SSTR2, SSTR3, and SSTR5. Combined with findings from the cell-based functional assays, the results of sequence alignment and structural analysis reveal that conserved residues 3.32-3.36-7.43 constitute a common motif for the pharmacophore of peptide agonist binding (Fig. 2l).

### Recognition mechanism of paltusotine by SSTR2

The small molecule paltusotine is a highly selective agonist of SSTR2 relative to other SSTR subtypes. The cryo-EM structure of the SSTR2-Gi complex bound to paltusotine showed that paltusotine occupies the bottom of the orthosteric binding pocket (OBP) (Figs. 2c, d and 3a). Notably, paltusotine largely resembles the core region of octreotide, with an aminopiperidin moiety and an aromatic ring of hydroxybenzonitrile mimicking Lys5 and (D)-Trp4, respectively, and makes extensive contacts with the major pocket of the OBP (Fig. 3b). In particular, the amide group of aminopiperidin forms a similar salt bridge with D122[3.32] or hydrogen bond with Y302[7.43] as that in the structure of the octreotide-bound receptor, whereas the hydroxybenzonitrile group of paltusotine is projected into a hydrophobic cavity that can

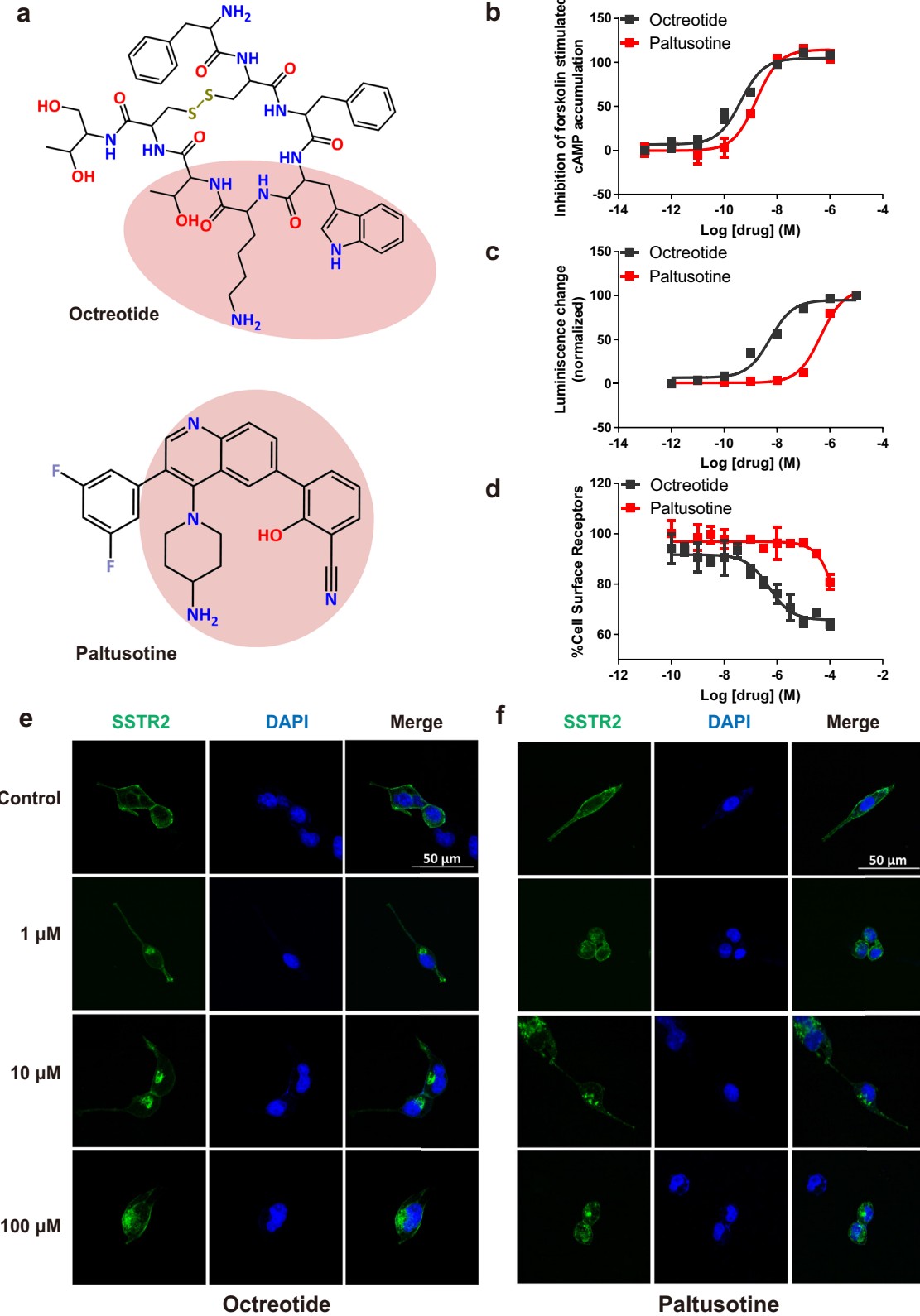

**Fig. 1 | Octreotide and paltusotine activate SSTR2 at the nanomolar level. a** The chemical structures of octreotide and paltusotine; similar moieties are highlighted in pink. **b**, **c** The efficacy of octreotide- and paltusotine-induced SSTR2 Gi signaling (**b**) and β-arrestin recruitment (**c**). Data represent mean ± SEM from three independent experiments. **d** Internalization of SSTR2 in HEK293 cells treated with octreotide and paltusotine measured by ELISA assays by detecting the expression of SSTR2 on the cell surface. Data represent mean ± SEM from three independent experiments. **e**, **f** eGFP-tagged SSTR2 plasmid was transfected in HEK293 cells and then cells were treated with octreotide (**e**) or paltusotine (**f**) for 30 min. The cells were analyzed by confocal fluorescence microscopy (green, eGFP-SSTR2; blue, DAPI). Scale bar, 50 μm.

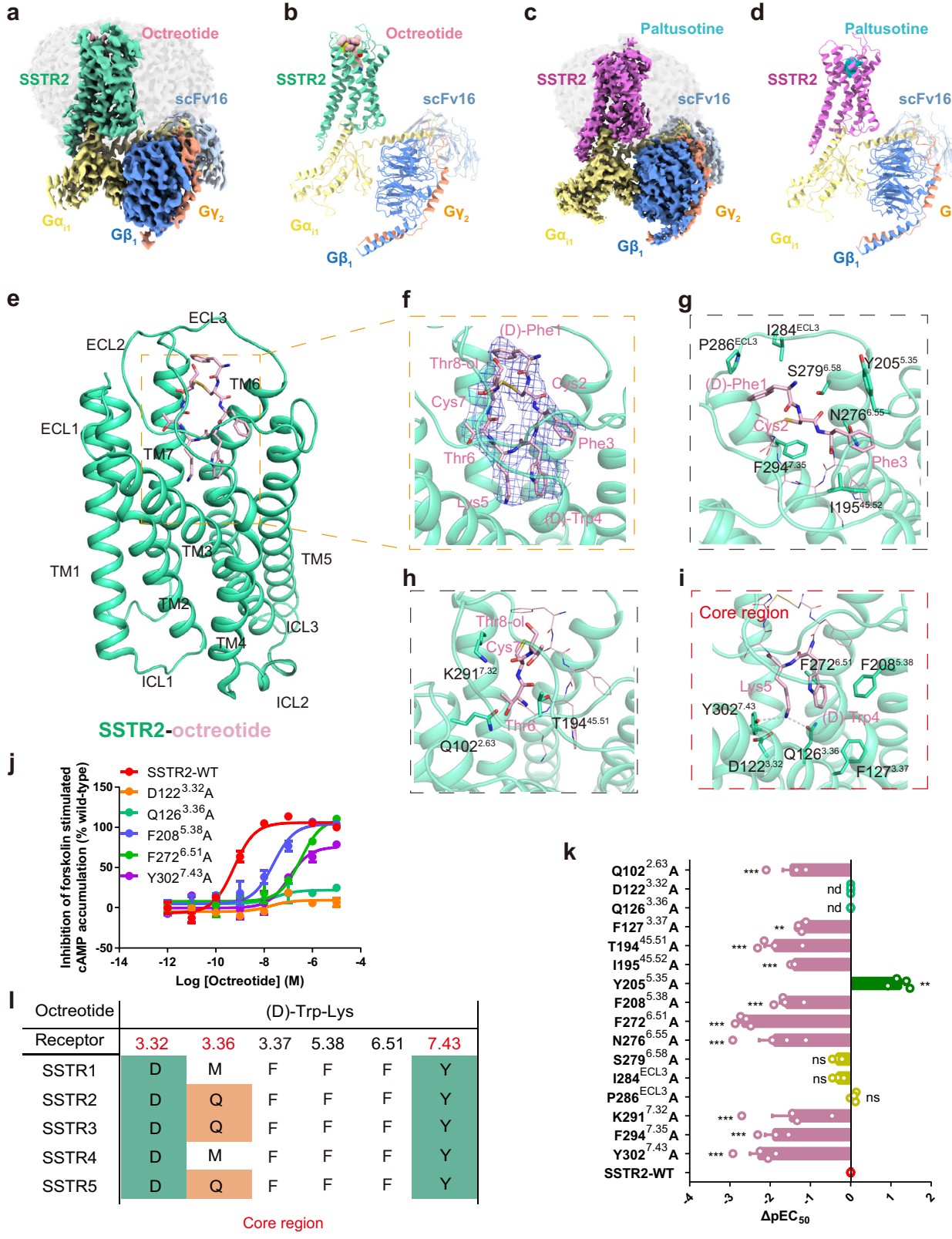

**SSTR2-octreotide**

| Octreotide | (D)-Trp-Lys | | | | |
|---|---|---|---|---|---|
| Receptor | **3.32** | **3.36** | 3.37 | 5.38 | 6.51 | **7.43** |
| SSTR1 | D | M | F | F | F | Y |
| SSTR2 | D | Q | F | F | F | Y |
| SSTR3 | D | Q | F | F | F | Y |
| SSTR4 | D | M | F | F | F | Y |
| SSTR5 | D | Q | F | F | F | Y |

Core region

accommodate the side chain of tryptophan from octreotide. The additional amine group makes a hydrogen bond with Q126[3.36] (Fig. 3c). We further performed mutagenesis studies and measured the cellular cAMP level induced by paltusotine using HEK293 cells expressing SSTR2. The results demonstrate that the common motif D[3.32]-Q[3.36]-Y[7.43] plays a critical role in diverse ligand recognition or receptor activation in SSTR2 (Fig. 3d and Supplementary Table 3).

A notable feature of paltusotine binding is that the large 3,5-difluorophenyl moiety is placed toward the extracellular end of TM2, creating a minor pocket formed by L99[2.60], Q102[2.63], and V103[2.64] in SSTR2. In addition, the residues Y50[1.39] and D295[7.36] from TM1 and TM7 are engaged with paltusotine recognition through van der Waals forces (Fig. 3e). The substitution of D295[7.36] with a bulkier side chain reduced the activation potency of the ligand (Supplementary Fig. 6).

**Fig. 2 | The binding mode of octreotide to SSTR2. a, b** The three-dimensional map (**a**) and overall structural model (**b**) of octreotide-bound SSTR2 in complex with Gi1 and scFv16. **c, d** The three-dimensional map (**c**) and overall structural model (**d**) of paltusotine-bound SSTR2 in complex with Gi1 and scFv16. **e, f** Structural details of octreotide bound to SSTR2. SSTR2 is shown in a cartoon diagram and colored in green-cyan; octreotide is shown as sticks and colored in light pink. Octreotide occupied the orthosteric binding pocket of SSTR2; the electron density map of octreotide is shown in tv-blue. **g–i** The detailed interaction network between octreotide and SSTR2. The detailed interaction network within (D)-Phe1-Cys2-Phe3 (**g**), Thr6-Cys7-Thr8-ol (**h**) and the core region (**i**) are shown. Amino acid residues involved in octreotide binding are shown as sticks. Hydrogen bonds are indicated as gray dashed lines. **j** The effects of SSTR2 mutations in the core region on octreotide-induced cAMP inhibition. Data represent mean ± SEM from three independent experiments for wild-type SSTRs and its mutants. WT: wild-type. **k** Effects of mutations involved in octreotide binding of SSTR2 on Gi signaling. Bars represent the $\Delta pEC_{50}$ relative to wild-type. Statistical differences between wild-type and mutants were determined by one-way of variance ANOVA with Dunnett's test. **$P < 0.02$; ***$P < 0.001$; ns, not significant; n.d., not detected ($P = < 0.001$, n.d., n.d., 0.002, <0.001, <0.001, 0.002, <0.001, <0.001, <0.001, 0.946, 0.961, >0.999, <0.001, <0.001, <0.001, from top to bottom). Data represent mean ± SEM from three independent experiments. **l** Sequence alignment of the residues around the core binding pocket of SSTR2 with other SSTR members.

## Comparison of binding modes of paltusotine and octreotide with SSTR2

The two structures of the SSTR2-Gi signaling complex bound to paltusotine or octreotide reveal a similar configuration of TM bundles on the extracellular side (Supplementary Fig. 6c). However, some significant differences are observed at the ligand binding pocket. Paltusotine lacks the extended region of octreotide (Fig. 3b), resulting in the loss of extensive contacts with the extracellular ends of TM5, TM6, ECL3, and TM7. Notably, the side chains of residues Y205[5.35], F294[7.35], T194[45.51], and I195[45.52] have notable displacements when the receptor senses paltusotine. Specifically, F294[7.35] slightly moves towards the difluorophenyl group of paltusotine, shortening the distance between T194[45.51] in ECL2 and F294[7.35] in TM7 by ~1.3 Å relative to the residue pair in octreotide-bound structure. The side chain of I195[45.52] projected further into the major pocket of the OBP, strengthening the contact with paltusotine. Moreover, the significant rearrangement of Y205[5.35] narrows the major pocket for paltusotine binding in SSTR2 (Fig. 3f). These observations from the structural comparison indicate different binding modes of SSTR2 with paltusotine and octreotide, and provide more evidence for the plasticity of GPCRs in response to diverse ligands.

Another difference lies in the common recognition motif (3.32-3.36-7.43) and classic microswitches in diverse ligand-bound structures. Compared with the amide group in the octreotide-bound receptor structure, the amide group of aminopiperidin in paltusotine is far away from Q126[3.36] but still retains direct contact with D122[3.32] and Y302[7.43] (Fig. 3g), altering the polar network important for receptor activation transition. Adjacent to the network, replacements of L96[2.57] and L99[2.60] in the minor pocket occur upon paltusotine binding, which subsequently drives the downstream benzene ring of F92[2.53] to flip down. Thus, these rearrangements cause obvious conformational change of the surrounding hydrophobic core of TM bundle in SSTR2 within the two complexes, particularly the microswitches W269[6.48] and P220[5.50]-I130[3.40]-F265[6.44] motif (Fig. 3h and Supplementary Fig. 6d).

## Selectivity of paltusotine for SSTR subtypes

The SSTR family comprises five members and is divided into two groups based on sequence homology, ligand binding characteristics, and pharmacological properties. SSTR1 and SSTR4 are classified into group 1, whereas SSTR2, SSTR3, and SSTR5 belong to group 2. Identification of selective ligands toward specific subtypes is challenging, especially for developing orally active small-molecule ligand. The selective mechanisms of octreotide and the small molecule L-054,264 for activating group 2 SSTRs over group 1 members have been reported[22,24]. The structure of SSTR2 in complex with paltusotine enabled the opportunity to examine ligand selectivity for distinguishing group 2 SSTRs, as paltusotine exerts higher selectivity and efficacy towards SSTR2 relative to other group 2 SSTRs (Fig. 4a and Supplementary Fig. 7a). In particular, L-054,264 is another small-molecule agonist of SSTR2 with distinct scaffold (Supplementary Fig. 7b), the residues F[7.35], F[6.54], and N[6.55] in TM6 and TM7 have been reported to participate in subtype selectivity of L-054,264 for activating group 2 SSTRs over group 1 members (SSTR1/4)[24], however, paltusotine lacks the direct contacts with the extended binding pocket involved in L-054,264 binding, due to different binding pose in SSTR2 (Supplementary Fig. 7c). Therefore, we tended to focus on the minor binding site engaged in paltusotine binding, sequence alignment reveals that the residues V103[2.64], T194[45.51], I195[45.52], Y205[5.35], F294[7.35], and D295[7.36] are divergent among group 2 SSTRs (Fig. 4b). Superposition of the active state of SSTR2 with SSTR3 (predicted active model from GPCRdb)[29,30] allowed us to inspect these differences in 3D structures (Supplementary Fig. 7d). We therefore speculate that these residues may play critical roles in the selectivity of paltusotine to SSTR2.

To further examine the role of the divergent residues in the activation of SSTRs, we examined cAMP accumulation using the cell-based GloSensor assay. The results revealed that T194[45.51] and V103[2.64] markedly affected the activation of SSTRs by paltusotine and octreotide. The T194[45.51]H substitution in SSTR2 significantly reduced receptor activation (Fig. 4c, Supplementary Fig. 7 and Supplementary Table 4); we speculate that the side chain of histidine caused steric hindrance, impairing the contact between both ligands with SSTR2. Additionally, the V103[2.64] residue in the minor pocket makes van der Waals forces contacts with the difluorophenyl moiety in paltusotine, and the V103[2.64]N mutation in SSTR2 reduced receptor activation induced by paltusotine. Both H192[45.51]T and N101[2.64]V substitutions in SSTR3 increased paltusotine-induced SSTR3 activation (Fig. 4d). In contrast, the V103[2.64]N mutant of SSTR2 and H192[45.51]T mutant of SSTR3 retained a similar $EC_{50}$ as wild-type receptor upon octreotide stimulation (Supplementary Fig. 7j, k and Supplementary Table 5). Together, these results reveal that the residues at positions 45.51 and 2.64 may not engaged in the selectivity of peptide agonist recognition, but they can determine subtype selectivity of paltusotine for SSTR2 among group 2 SSTRs.

## Signal bias properties of SSTR2 with different ligands

Paltusotine induced lower β-arrestin recruitment compared with octreotide (Fig. 1c and Supplementary Fig. 8a). To further investigate the signal bias of paltusotine and octreotide, we performed a comprehensive comparison to identify the divergent contacts in the two ligand-bound structures. An extended region of octreotide interacts with the extracellular ends of TM6/7 and ECL3, whereas paltusotine is placed into a minor pocket in SSTR2. We next explored whether the corresponding contact residues were involved in signal bias of the receptor.

We first generated a range of mutations of the residues within the extended binding pocket occupied by octreotide and minor pocket by paltusotine; substitution of I284[ECL3] and K291[7.32] with alanine resulted in reduced potency and efficacy of octreotide induced β-arrestin signal, while the impact on the Gi protein pathway was relatively small (Supplementary Fig. 8b and Supplementary Table 2). Notably, the N276[6.55]A mutation of SSTR2 nearly abolished the recruitment of β-arrestin, but the receptor still exerted Gi signaling activation induced by octreotide (Fig. 5a, b and Supplementary Table 2). By contrast, paltusotine loses the interaction with the extended region, thus the I284[ECL3]A, K291[7.32]A, and N276[6.55]A mutants of SSTR2 exhibited slightly enhanced recruitment of β-arrestin when activated by paltusotine (Fig. 5c, d,

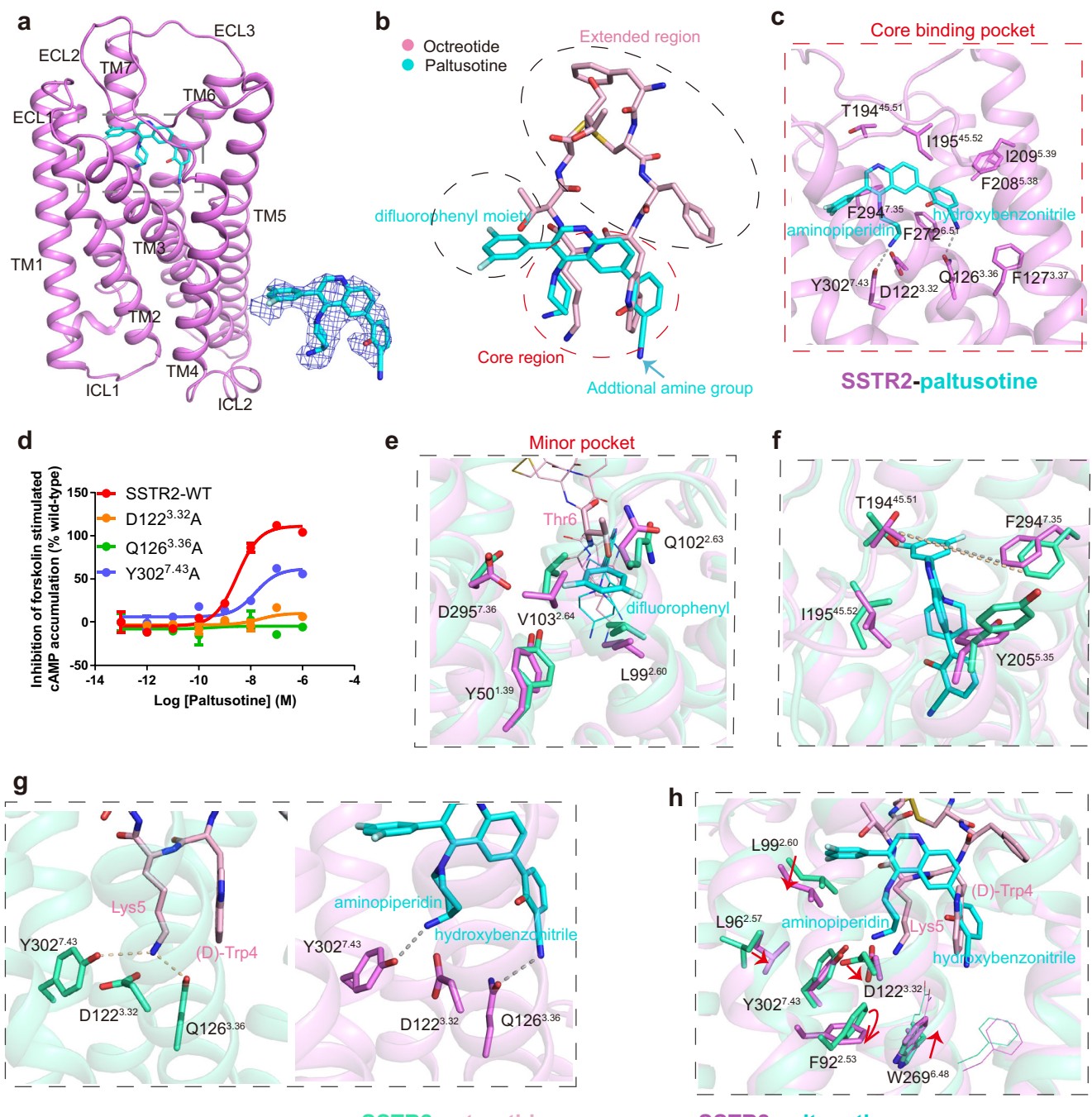

**Fig. 3 | The differential binding mode of paltusotine by SSTR2. a** Structural details of paltusotine bound to SSTR2. SSTR2 is shown in cartoon diagram and colored in violet; paltusotine is shown as sticks and colored in cyan. Paltusotine occupied the bottom of the orthosteric binding pocket of SSTR2; the electron density map of paltusotine is shown in tv-blue. **b** Three-dimensional structural comparison of octreotide (light pink) and paltusotine (cyan). **c** The detailed interaction network between paltusotine and the SSTR2 core binding pocket. Amino acid residues involved in paltusotine binding are shown as sticks. Hydrogen bonds are indicated as gray dashed lines. **d** The effects of the SSTR2 mutations in the core region on paltusotine-induced cAMP inhibition. Data represent mean ± SEM from three independent experiments for wild-type SSTR2 and its mutants. WT: wild-type. **e** The detailed interaction network within the paltusotine-binding minor pocket. Amino acid residues within the minor pocket are shown as sticks and colored in green-cyan (octreotide-bound SSTR2) and violet (paltusotine-bound SSTR2). **f–h** Conformational comparison of residues involved in divergent parts of ligand binding are shown in **f** (T194[45.51] and F294[7.35] residue pair), **g** (core region) and **h** (downstream). Key residues are shown in sticks and colored in green-cyan (octreotide-bound SSTR2) and violet (paltusotine-bound SSTR2); octreotide is colored in light pink and paltusotine is colored in cyan.

Supplementary Fig. 8b and Supplementary Table 3). In addition, residue F294[7.35] in the orthosteric binding pocket forms hydrophobic interaction with the disulfide bond of octreotide, the side chain of F294[7.35] is closer to octreotide than paltusotine. Alanine replacement of F294[7.35] nearly impaired the octreotide-induced β-arrestin recruitment

(Fig. 5e, f), yet slightly influenced the β-arrestin pathway activated by paltusotine (Fig. 5g, h and Supplementary Fig. 8b). These results suggest that N276[6.55], I284[ECL3], K291[7.32]A, and F294[7.35] may be key residues that contribute to the signal bias of SSTR2 when sensing octreotide. On the other hand, V103[2.64] from minor pocket makes van der Waals forces

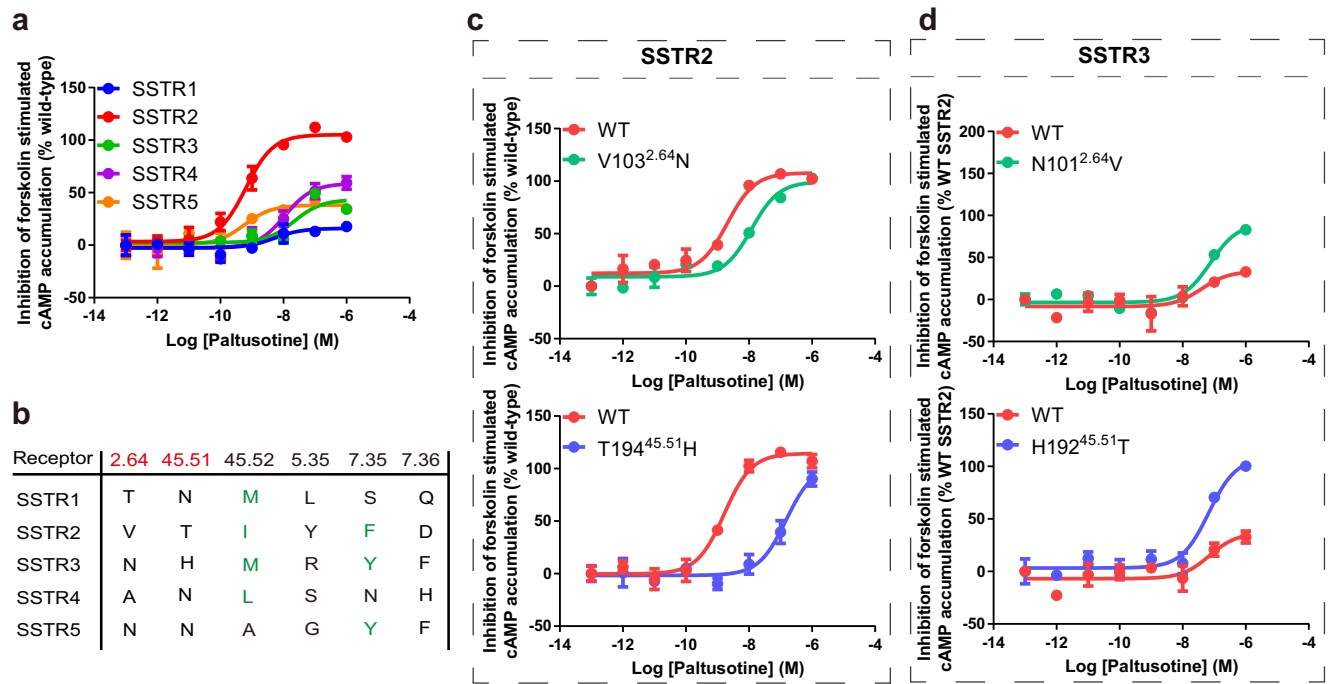

**Fig. 4 | The molecular mechanism of the SSTR subtype selectivity by paltusotine over octreotide. a** Gi signaling of SSTR1–5 activated by paltusotine. Data represent mean ± SEM from three independent experiments. **b** Sequence alignment of the residues around the paltusotine binding pocket in SSTR2 with sequences of other SSTR subtypes. **c**, **d** The effects of the residue substitution of N103$^{2.64}$V or T194$^{45.51}$H in SSTR2 (**c**) and V101$^{2.64}$N or H192$^{45.51}$T in SSTR3 (**d**) on the paltusotine-induced cAMP inhibition. Data represent mean ± SEM from three independent experiments.

with paltusotine, alanine substitution of V103$^{2.64}$ markedly reduced the paltusotine-induced β-arrestin signal while had less impact on Gi signaling (Supplementary Fig. 8c and Supplementary Tables 2, 3), which indicates that avoiding the interaction with V103$^{2.64}$ may further reduce β-arrestin recruitment.

In addition, we measured the function of the residues located in the core region and the common ligand binding pocket. The Y302$^{7.43}$A substitution resulted in markedly decreased G-protein pathway activity, but it retained β-arrestin recruitment, while most of the mutants (e.g., D122$^{3.32}$ and Q126$^{3.36}$) markedly affected both β-arrestin and Gi signal under the treatment of whether octreotide or paltusotine. Residues like Q102$^{2.63}$A mainly influenced the β-arrestin recruitment but without ligand dependence. Such residues were observed to participate in signal bias regulation, which is presumably a general feature of the receptor (Supplementary Fig. 8c and Supplementary Tables 2, 3). Moreover, we carried out the β-arrestin recruitment assays with co-transfection of GRK2, no significant differences for the efficacy of ligand-induced β-arrestin recruitment were observed in the presence or absence of GRK2[16,31].

The class A peptide receptors angiotensin II type 1 receptor (AT1R) and opioid receptor μOR are prototypical GPCRs that have been studied for their ligand binding and signal bias features. We found that the ligand-binding pattern of SSTR2 is distinct from those reported GPCR complex structures. In SSTR2, octreotide or paltusotine locates in a relative shallow orthosteric binding pocket. In μOR, the ligands BU72 and FH210 insert deeply into the ligand binding pockets, making direct interactions with the W$^{6.48}$ microswitch residue[32,33]. Accordingly, the AT1R arrestin-biased ligand TRV023 promoted the rearrangement of the residues within the bottom of the binding site[34] (Supplementary Fig. 8d). We wondered whether these residues located at the bottom of the orthosteric pocket of SSTR2 were involved in the biased signal transition. Therefore, we conducted mutagenesis in the core of the TM bundle, especially in critical microswitch residues that are required for receptor activation propagation, and tested the effects on the G-protein and β-arrestin signal pathway. Mutations of the conserved microswitches C$^{6.47}$W$^{6.48}$xP$^{5.50}$, P$^{5.50}$I$^{3.40}$F$^{6.44}$, D$^{3.49}$R$^{3.50}$Y$^{3.51}$, and N$^{7.49}$P$^{7.50}$xxY$^{7.53}$ motifs decreased the potency of β-arrestin recruitment more markedly than the activation ability of Gi signaling compared with wild-type SSTR2 when sensing both octreotide and paltusotine, and the microswitch residues involved signal bias property is not ligand dependent (Fig. 5i–k).

## Influence of SSTR2 internalization by the ligand-induced β-arrestin signal

We further explored the impact of the residues affecting the β-arrestin signal on the internalization of SSTR2. Wild-type SSTR2 was obviously internalized in response to octreotide, leading to desensitization, by contrast, internalization and endosomal trafficking of SSTR2 were reduced under paltusotine treatment at the same concentration (Fig. 1d–f and Supplementary Fig. 9a).

Consistent with our cell-based signaling experiments, the results of ELISA assay, Bystander BRET based endosomal trafficking assay using FYVE, an early endosome marker[35,36] or confocal assay reveal that SSTR2 mutants bearing I284$^{ECL3}$, K291$^{7.32}$A, N276$^{6.55}$A, and F294$^{7.35}$A substitutions showed diminished internalization when treated with octreotide, while exhibited slightly change in the internalization under paltusotine treatment (Fig. 6, Supplementary Fig. 9b–c, d, g, j, k and Supplementary Tables 6–7). Different from these residues, alanine substitution of the microswitch residues mentioned above all resulted in reduced receptor internalization when sensing both octreotide and paltusotine (Supplementary Fig. 9e, f, h, i). Therefore, our results suggest that these residues should contribute to the octreotide-induced β-arrestin signal and the internalization of SSTR2.

Collectively, we mapped the signaling bias–related residues in the structure and found that within the octreotide extended region, I284$^{ECL3}$, K291$^{7.32}$, N276$^{6.55}$, and F294$^{7.35}$ in the extended region of octreotide contributed to activation of the β-arrestin signal and the internalization of SSTR2 activated by octreotide. Whereas the downstream microswitch residues involved in the efficacy of β-arrestin recruitment induced by both ligands (Fig. 5k).

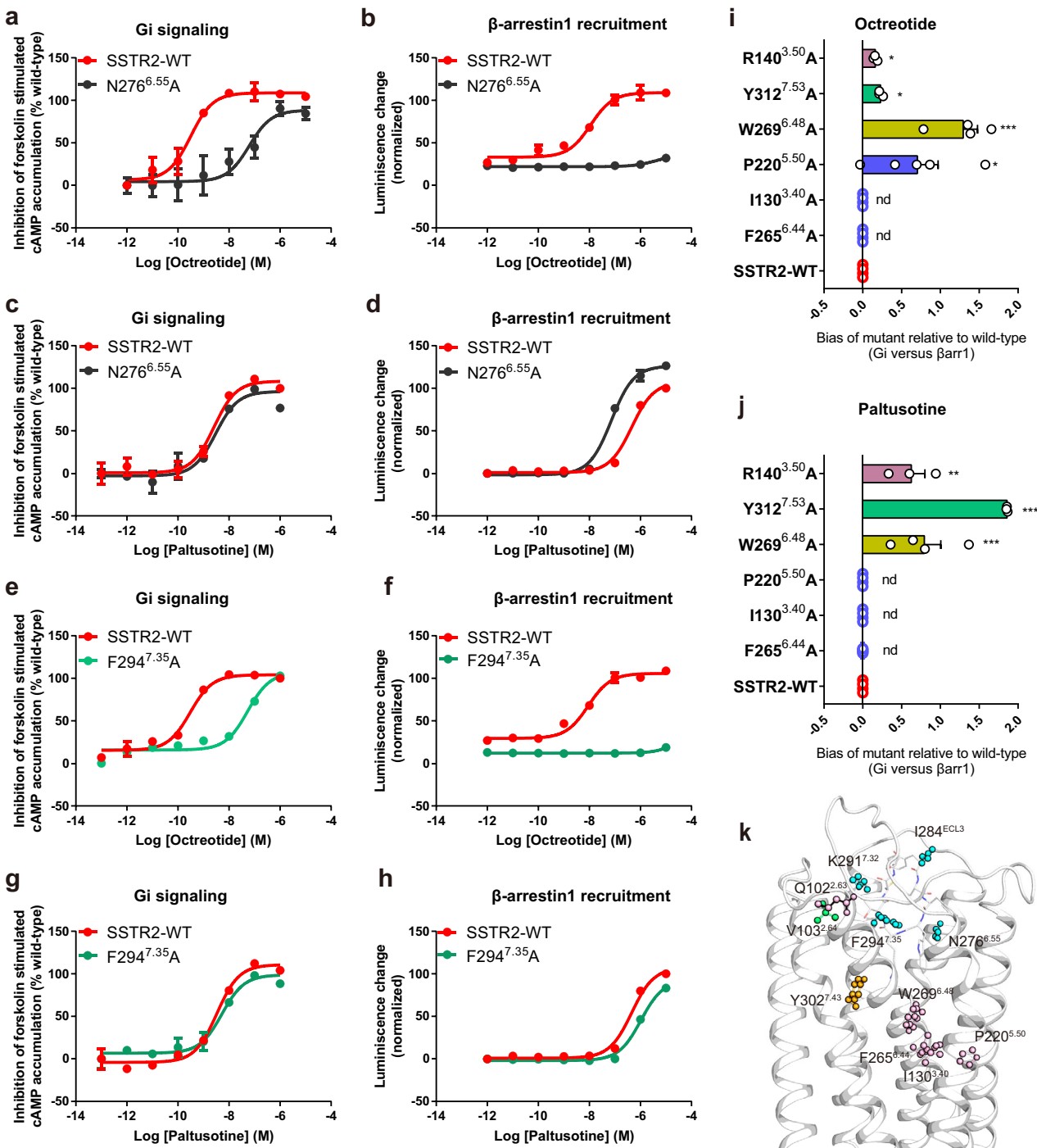

**Fig. 5 | Signal bias of SSTR2 induced by octreotide and paltusotine. a, b** The effects of SSTR2 N276⁶·⁵⁵A mutation on cAMP inhibition (**a**) and β-arrestin recruitment (**b**) induced by octreotide. Data represent mean ± SEM from three independent experiments. **c, d** The effects of SSTR2 N276⁶·⁵⁵A mutation on cAMP inhibition (**c**) and β-arrestin recruitment (**d**) induced by paltusotine. Data represent mean ± SEM from three independent experiments. **e, f** The effects of SSTR2 F294⁷·³⁵A mutation on cAMP inhibition (**e**) and β-arrestin recruitment (**f**) induced by octreotide. Data represent mean ± SEM from three independent experiments. **g, h** The effects of SSTR2 F294⁷·³⁵A mutation on cAMP inhibition on cAMP inhibition (**g**) and β-arrestin recruitment (**h**) induced by paltusotine. Data represent mean ± SEM from three independent experiments. **i, j** Bias factors of the substitution of the microswitch residues in SSTR2 relative to wild-type induced by octreotide (**i**) and

paltusotine (**j**). Statistical differences between wild-type and mutants were determined by one-way of variance ANOVA with Dunnett's test. *$P < 0.033$, **$P < 0.02$; ***$P < 0.001$; n.d., not detected. Octreotide ($P = 0.022$, 0.012, <0.001, 0.012, n.d., n.d., from top to bottom). Paltusotine ($P = 0.002$, <0.001, <0.001, n.d., n.d., n.d., from top to bottom). Data represent mean ± SEM from three independent experiments. **k** Residues that markedly affect β-arrestin signaling of SSTR2 are shown in sphere sticks. Residues involved in both octreotide- and paltusotine-induced arrestin recruitment are colored in light pink; residues mainly affect the octreotide-induced β-arrestin recruitment are colored in cyan; residues mainly affect the paltusotine-induced β-arrestin recruitment are colored in lime green; residues that mainly affect Gi protein signaling are colored in bright orange.

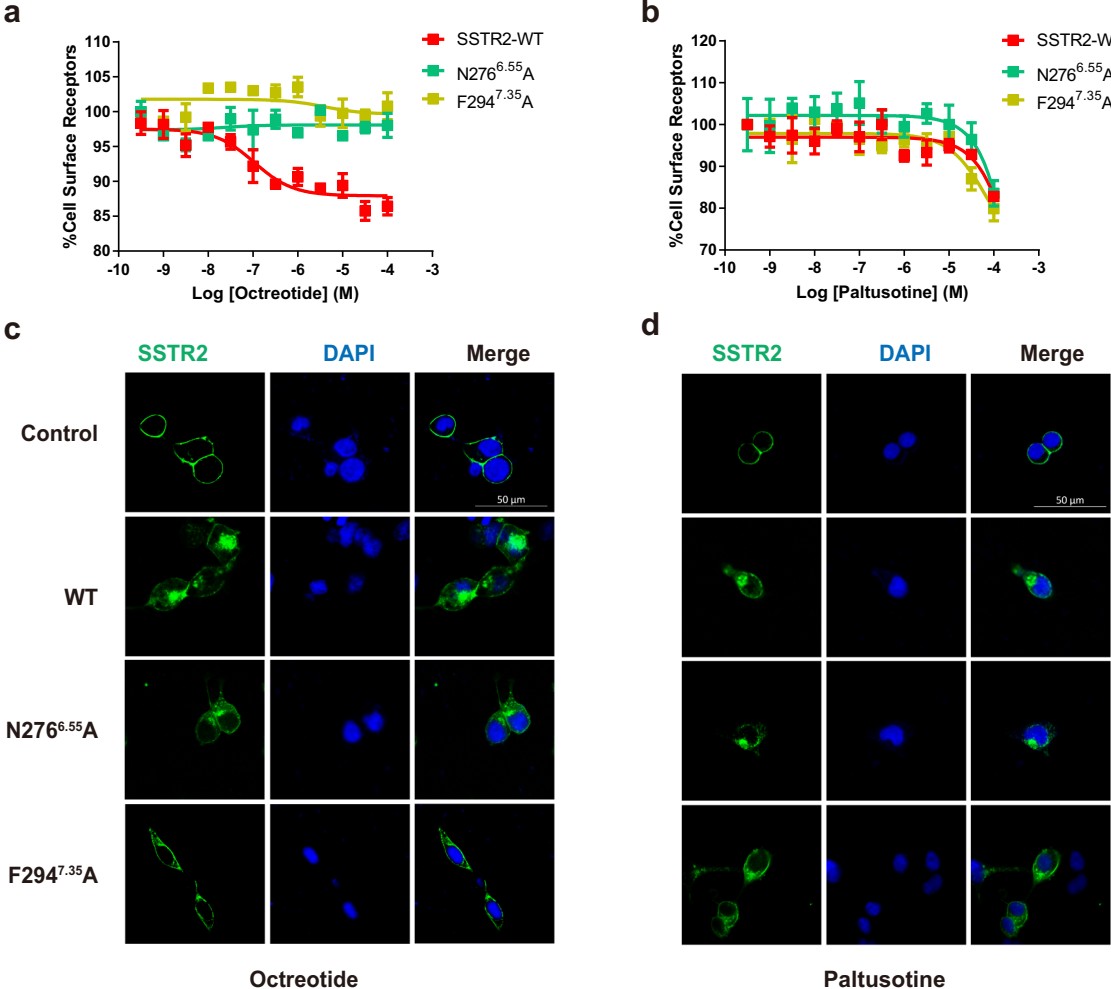

**Fig. 6 | Differential internalization activated by octreotide and paltusotine.**
**a**, **b** Effects of N276[6.55]A or F294[7.35]A mutation on the internalization of SSTR2 in
HEK293 cells treated with octreotide (**a**) and paltusotine (**b**) measured by ELISA
assays by detecting the expression of SSTR2 on the cell surface. Data represent
mean ± SEM from three independent experiments. **c**, **d** Effects of N276[6.55]A or
F294[7.35]A mutation on the internalization of SSTR2 in HEK293 cells treated with
octreotide (**c**) and paltusotine (**d**) indicated by confocal fluorescence microscopy.
eGFP-tagged SSTR2 wild-type or mutant plasmid was transfected in HEK293 cells
and then cells were treated with 10 μM octreotide or paltusotine for 30 min and
analyzed by confocal fluorescence microscopy (green, eGFP-SSTR2; blue, DAPI).
Scale bar, 50 μm.

## Octreotide and paltusotine promote apoptosis of pituitary tumor cell

Treatment of NETs by targeting SSTR2 signaling pathway was previously demonstrated to induce hormone secretion and tumor apoptosis[37,38]. Furthermore, we inferred whether small molecule paltusotine or peptide drug octreotide play a role in apoptosis process on pituitary tumor cell. Subsequently, the apoptotic rate of the rat pituitary-derived GH3 cells that overexpress SSTR2 was measured by flow cytometry Annexin V-FITC/PI analysis. The result reveals that both octreotide and paltusotine are able to activate proapoptotic pathways. Notably, increased apoptotic rate was observed when treating with paltusotine over octreotide (Fig. 7a, b and Supplementary Fig. 10), consistent with the bias profile of clinical drugs, the evidence also indicates a potential better performance of the small-molecule drug in clinical treatment of NETs through promoting tumor apoptosis.

## Discussion

GPCRs regulate a broad range of physiological and pathological processes by triggering different signaling pathways, including G-protein coupling and arrestin recruitment. Previous study has indicated that one of the receptor signaling pathways may be responsible for the therapeutic effects, whereas the other mediates unwanted side effects[39]. As most GPCR subfamilies, SSTRs distributed in different tissues and regulated divergent physiological processes, therefore, designing selective ligands that can achieve receptor subtype selectivity and specific signaling and even control on- or off-target side effects, could be safer therapeutic strategy. SSTR2 signals via activation of G-protein and engages β-arrestin to mediate distinct cellular signaling events and has been proved to be a valuable drug target for the treatment of acromegaly, however, subtype selectivity of SSTRs and signal bias properties of different types of ligands (peptide octreotide and small molecule paltusotine) hinder the development of effective drugs targeting SSTR2. Our study, to a certain extent, contributes to the understanding of the functional bias of ligands and guiding rational drug design targeting SSTRs.

The small-molecule agent paltusotine undergoing phase 3 trials for the treatment of NETs acts as a highly selective agonist of SSTR2, herein, we investigated the recognition mechanism by determining complex structure of SSTR2 bound to paltusotine, and an unusual minor pocket in SSTR2 was demonstrated to play an important role in ligand selectivity. In particular, the substitution of T194[45.51] and V103[2.64] in SSTR2 with corresponding residues (H192[45.51] and V103[2.64]) in SSTR3 reduced receptor activation induced by paltusotine (Fig. 4c and Supplementary Fig. 7). Moreover, T194[45.51] interacted with Thr6 in

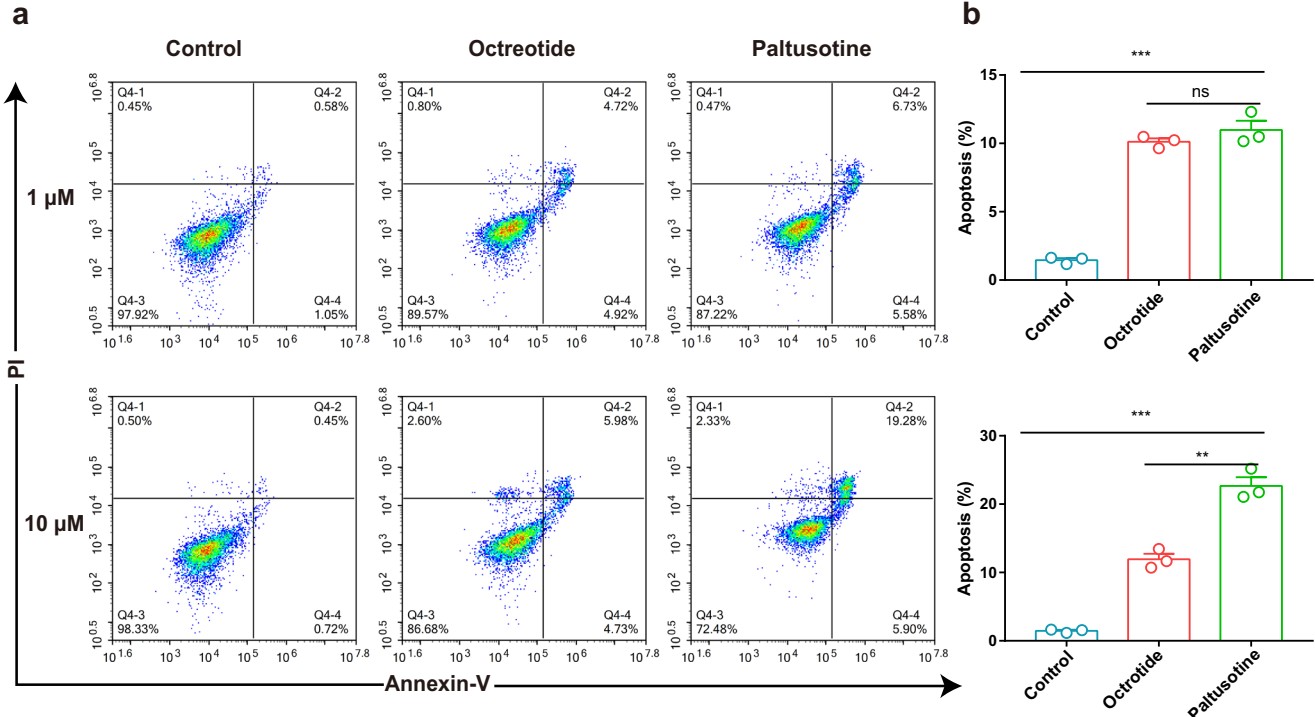

**Fig. 7 | Octreotide and paltusotine promote pituitary tumor cell apoptosis.**
**a**, **b** Statistic results of GH3 cell apoptosis assays of control, octreotide or paltusotine treated cells. Cells for Annexin V+/PI− and Annexin V+/PI+ were both considered apoptotic (**a**). Bars represent the percentage of apoptotic cells in **a**.
**b** Statistical differences between control and octreotide or paltusotine treated cells were determined by one-way of variance ANOVA with T-test. Data are expressed as mean ± SEM from three independent experiments (**P < 0.02; ***P < 0.001; n.s., not significant). (Control compared with octreotide or paltusotine (P = < 0.001 under the treatment of 1 μM drug; P = < 0.001 under the treatment of 10 μM drug). Octreotide compared with paltusotine (ns, under the treatment of 1 μM drug; P = < 0.02 under the treatment of 10 μM drug)). The gating strategy of apoptotic cells was provided in Supplementary Fig. 10.

octreotide, and it was replaced by histidine or asparagine in SSTR3 or R5, which might be responsible for the higher octreotide efficacy for SSTR2 (Supplementary Fig. 7j) and reduced affinity for SSTR3/5. During the preparation of our manuscript, several individual groups reported the structures of SSTR2 bound to octreotide and other different types of ligands[21–25]. Subtype selectivity of peptide ligands at SSTR2 is multifaceted because of extensive contacts, especially divergent ECL regions in SSTRs which were suggested to participate in the selectivity of peptide agonist[21]. Together with previous reporting, our findings further profile the subtype selectivity of ligand to group 2 SSTRs (SSTR2/3/5) over group 1 SSTRs (SSTR1/4). Emerging of the structures and pharmacological studies of SSTRs will be helpful to understand subtype selective mechanism and paves a way to design effective small molecules toward subtype receptor.

Moreover, clinical studies have shown that lower expression of β-arrestin 1 in acromegalic patients may be associated with decreasing recycling rate of SSTR2 and better SST analog response[20,40]. Understanding of the bias properties of SSTR2 would help pharmacologists to design or optimize small molecules with biased signaling. Interestingly, we characterized that paltusotine acts as more G protein-biased ligand than octreotide, and provided structural insights to understand the bias property of paltusotine. In the paltusotine-bound structure, we found that paltusotine loses the interactions with residues I284[ECL3], K291[7.32], N276[6.55], and F294[7.35] that participated in the recruitment of β-arrestin induced by octreotide (Fig. 5, Supplementary Figs. 8 and 9). Moreover, V103[2.64] contributed to the β-arrestin recruitment under the treatment of paltusotine. Designing specific drugs target the core region and with minimum contact with the extended binding pocket of SSTR2 would facilitate the development of G protein-biased ligand of SSTR2. Future studies would be highly informative to solve the SSTR2-β-arrestin complex structure for precise pharmacological

characterization of novel drugs targeting SSTRs. Collectively, designing selective ligands or modulators that control specific subtype and functional signaling could be safer therapeutic strategy in GPCR drug discovery field.

## Methods
### SSTRs Gi-mediated cAMP inhibition assay
The levels of the intracellular 3′, 5′-cyclic AMP (cAMP) were measured by the GloSensor cAMP assay (Promega) as described in the previous studies[28,41] which has been widely used. The human full-length SSTRs and mutants were cloned into the pcDNA3.1+ vector with a HA signal sequence followed by a Flag tag at the N-terminus. HEK293 cells (ATCC, CRL-1573) were seeded into 6-well microplates at a density of $1.0 \times 10^6$ cells/well and cultured for 24 h at 37 °C with 5% $CO_2$. Cells were then co-transfected with SSTR wild-type or mutants and GloSensor plasmids. After 24 h of transfection, the cells were dissociated with TrypLE™ Express Enzyme (1X) (Thermo Fisher Scientific) and collected by centrifugation at $500 \times g$ for 5 min. Then the collected cells were suspended in Hank's balanced salt solution [HBSS (Thermo Fisher Scientific)] supplemented with D-fluorescein potassium salt solution (YEASEN), and then the suspension cells were seeded into 96-well plates with 90 μl/well. After incubating at room temperature for 40 min, 5 μM forskolin and ligands diluted in HBSS were added to each well and the plates were incubated at room temperature for 15 min. Fluorescence signals were measured using the Synergy H1 microplate reader (BioTek) and the data presented are means of at least three independent experiments.

### NanoBiT β-arrestin 1 recruitment assay
The NanoBiT β-arrestin 1 recruitment assays were carried out according to the method described previously[42]. Human full-length β-arrestin

1 was cloned into pcDNA3.1+ vector with the large fragment (LgBiT) of the NanoBiT system and a 15-amino acid flexible linker (GGSGGGGSGGSSSGG) at the N-terminus. Human full-length SSTR wild-type and mutant vectors were C-terminally fused with a 15-amino acid flexible linker and the small fragment (SmBiT). HEK293 cells were seeded into six-well microplates at a density of $1.0 \times 10^6$ cells/well and cultured for 24 h at 37 °C with 5% $CO_2$. After that, cells were co-transfected with pcDNA3.1+ vectors encoding SSTR wild-type or mutants and human β-arrestin 1. After transfecting for 24 h, the transfected cells were suspended and seeded into white 96-well plates, and then cultured at 37 °C overnight. After that the medium in each well was exchanged for 90 μl of 0.01% bovine serum albumin containing 20 μl of 50 μM coelenterazine h (Carbosynth) diluted in the HBSS assay buffer and incubated for 40 min at room temperature. Then, ligands diluted in the assay buffer was added to each well and the luminescence intensity was measured using the Synergy H1 microplate reader and detected at room temperature in a time period. All data presented are means of at least three independent experiments.

### Detection of surface expression of SSTR wild-type and mutants

HEK293 cells were seeded into 96-well microplates at a density of $4 \times 10^4$ cells/well and cultured for 24 h at 37 °C with 5% $CO_2$ and were transfected using the same procedure as the cAMP accumulation assay. The plates were washed with washing buffer (PBS) the day after transfection, and were fixed with 4% formalin at room temperature for 30 min. Cells were washed with washing buffer and blocked with PBS supplemented with 5% BSA for 1 h at room temperature. After that, cells were incubated with anti-Flag antibody (1:1000 diluted with PBS containing 5% BSA, Proteintech: 66008-4-Ig) for 1 h at room temperature. After 1 h incubation, cells were washed with washing buffer and incubating with HRP-conjugated secondary antibody for 2 h at room temperature. After incubating and washing, 50 μl of chromogenic solution was added to each well, and reacted at room temperature for 10 min, and then 0.5 M HCl were added to stop the reaction. Optical density (OD) was measured at 455 nm in Synergy H1 microplate reader and all data presented are means at least three independent experiments.

### Quantification of SSTR2 internalization by ELISA assay

The quantification of SSTR2 internalization was performed following the procedure of the "Detection of surface expression of SSTR wild-type and mutants" as described in the last paragraph[41], except that octreotide (MCE, HY-17365) or paltusotine (MCE, HY-109155) need to be added and incubated at 37 °C for 30 min before being fixed with 4% formalin. The absorbance at 455 nm was measured using Synergy H1 microplate reader and was compared with untreated cells. All data presented are means at least three independent experiments.

### Internalization confocal assay

The internalization assay was performed as previously reported[43]. Briefly, HEK293 cells were transfected with plasmids encoding eGFP-tagged wild-type or mutant SSTR2. The transfected cells were plated on poly-D-lysine (Beyotime) coated coverslips in 24-well plates and grew overnight. The cells were washed with PBS, and then octreotide or paltusotine was added at the final concentration of 1–100 μM. After incubation for 30 min at 37 °C, the cells were fixed and permeabilized with methanol (−20 °C) for 5–10 min. Finally, the coverslips were mounted onto glass slides with the antifade mounting medium containing DAPI (Beyotime Biotechnology, P0131-5 ml). The confocal fluorescence images were aquired by Zeiss LSM 880 microscope with the ZEN imaging software (Zeiss).

### Cell apoptosis assay

The rat pituitary GH3 cells (ATCC, CCL-82.1) were seeded in 12-well plates at $1.5 \times 10^5$/well. Octreotide or paltusotine were added to each well at the indicated concentrations, and incubated for 48 hours. Then, the cells were collected for the apoptosis assay according to the instruction of the Annexin V-FITC/PI apoptosis detection kit (BD Biosciences). Briefly, the cells were washed with cold PBS and resuspended in 1 × binding buffer. Then, 5 μl of Annexin V-FITC was added and incubated for 10 min and stained with 5 μl of PI for 5 min in the dark at room temperature. All samples were acquired by a Novocyte flow cytometer (ACEA Biosciences, China), and data were analyzed with NovoExpress™ software.

### Constructs for structure determination

For the octreotide-bound SSTR2-Gi1 complex, the full-length of wild-type human SSTR2 ($SSTR2^{FL}$) cDNA (UniProt: P30874) was cloned into the pFastBac1 vector with the haemagglutinin (HA) signal sequence and a Flag epitope tag at the N-terminus. To purify the paltusotine-bound SSTR2-Gi1 complex, the amino acid residues 1–30 at the N-terminus and residues 339–369 at the C-terminus of SSTR2 were truncated ($SSTR2^{31-338}$). The wild-type human $G\alpha_{i1}$ (UniProt: P63096) without any modification ($G\alpha_{i1}^{WT}$) and the human $G\alpha_{i1}$ with four dominant-negative mutations ($G\alpha_{i1}^{4M}$; S47N, G203A, E245A, A326S) were subcloned into pFastBac1 vector[44]. The human $G\beta_1$ containing an N-terminal 10 x His tag and the human $G\gamma_2$ were cloned into a pFastBac-dual vector for co-expression.

### Expression and purification of SSTR2-Gi complex

The octreotide-bound SSTR2-Gi complex was expressed by co-infecting the HA-Flag-$SSTR2^{FL}$, $G\alpha_{i1}^{WT}$, and $G\beta_1G\gamma_2$ baculovirus into *Spodoptera frugiperda* (Sf9, Expression system, 94011 S) cells at a density of $3 \times 10^6$ cells/ml. After 56 h, cells were collected and lysed with 20 mM Hepes (pH 7.5), 50 mM NaCl, 5 mM $MgCl_2$, 5 mM $CaCl_2$, Protease Inhibitor Cocktail (160 μg/ml benzamidine, 100 μg/ml leupeptin), and 25 mU/ml Apyrase (NEB) and 10 μM octreotide at room temperature for 2 h. Then the membrane fractions were solubilized with 20 mM Hepes (pH 7.5), 100 mM NaCl, 5 mM $MgCl_2$, 5 mM $CaCl_2$, 0.5% (w/v) lauryl maltose neopentyl glycol (LMNG; Anatrace), 0.1% (w/v) cholesteryl hemisuccinate (CHS), 10% (v/v) glycerol, Protease Inhibitor Cocktail, 10 μM octreotide, 0.02 mg/ml scFv16, and 25 mU/ml Apyrase (NEB) followed by dounce homogenization and incubated for 2 h at 4 °C. After centrifuging at $65,000 \times g$ for 30 min, the solubilized complex was incubated with M1 anti-Flag antibody coupled Sepharose Resin at 4 °C for 2 h. Then the target complex was eluted with 20 mM Hepes (pH 7.5), 100 mM NaCl, 0.01% (w/v) LMNG, 0.002% (w/v) CHS, 10 mM EDTA, 0.2 mg/ml Flag peptide and 10 μM octreotide. The concentrated sample was loaded onto a Superose6 increase 10/300 size exclusion column (GE Healthcare) in the buffer include 20 mM Hepes (pH 7.5), 100 mM NaCl, 0.00075% (w/v) LMNG, 0.00025% glyco-diosgenin (GDN; Anatrace), 0.0002% (w/v) CHS and 5 μM octreotide. Target complex was collected and concentrated to 5 mg/ml using an Amicon Ultra Centrifugal Filter (MWCO, 100 kDa cut-off). The paltusotine-bound SSTR2-Gi complex was prepared using the same protocol, except that SSTR2 fragment ($SSTR2^{31-338}$), and $G\alpha_{i1}$ with four dominant-negative mutations ($G\alpha_{i1}^{4M}$) were used, and paltusotine was used throughout the buffers instead of octreotide.

### Expression and purification of scFv16

The procedures of the expression and purification were performed as previously described[45]. Briefly, scFv16 was cloned into pfastBac1 with the C-terminal 6 x His tag and the N-terminal GP67 signal peptide. Protein was expressed using the Sf9 baculovirus system and purified by Ni-NTA resin. The eluted protein was further concentrated to 4 mg/ml in a buffer contain 20 mM Hepes (pH 7.5) and 100 mM NaCl. Then the protein was flash frozen in liquid nitrogen and was stored at −80 °C for further use.

## Cryo-grid preparation and EM data collection

3 µl of the octreotide-bound SSTR2-Gi1 complex at ~5 mg/ml or the paltusotine-bound SSTR2-Gi1 complex at ~20 mg/ml was applied onto glow-discharged 300-mesh Au grids (Quantifoil R1.2/1.3), respectively. The grids were blotted for 2–3 s before swiftly frozen in liquid ethane using a Vitrobot Mark IV (Thermo Fisher) at 4 °C and 100% humidity. The octreotide-bound SSTR2-Gi complex was loaded into a Titan Krios cryo-electron microscope (Thermo Fisher) with a Gatan K3 direct electron detector, data collection was performed at 300 keV with the magnification of 130,000 (corresponding to a calibrated sampling of 0.46 Å per physical pixel). At a dose rate of 20 e⁻/pix/s and a total exposure period of 2.7 s, the SerialEM software[46] was used to automatically gather movie stacks in super resolution mode, providing 40 frames per stack and a total dose of 65 e⁻/Å². While the paltusotine-bound SSTR2-Gi-scFv16 complex sample was loaded into a Titan Krios Cryo-EM (Thermo Fisher) with a Gatan K2 direct electron detector, data collection was performed at 300 keV with the magnification of 165,000 (corresponding to a calibrated sampling of 0.85 Å per physical pixel). With a dose rate of 7.8 e⁻/pix/s and a total exposure period of 6 s, the EPU software (Thermo Fisher) was used to automatically gather movie stacks in counting mode, producing 36 frames per stack and a total dose of 65 e⁻/Å².

## Image processing and 3D reconstructions

6157 and 4746 movie stacks were obtained for the octreotide-bound SSTR2-Gi1 complex and paltusotine-bound SSTR2-Gi1 complex, respectively. The movie stacks were motion corrected using Motioncor2[47]. Contrast transfer function (CTF) parameters were estimated by GCTF[48] and Micrographs with a resolution <4 Å were eliminated. For the octreotide-bound SSTR2-Gi1 complex, a total of 2,364,681 particles were auto-picked and extracted in Relion 3.0[49] with the box size of 256 pixels. Following two rounds of 2D classifications, bad particles were removed and 1,159,274 well-defined particles were subjected to cryoSPARC3.1[50] ab initio to build the initial model. The resulting particles were then subjected to a round of cryoSPARC heterogeneous refinement. The best class determined by visual examination was subjected to cryoSPARC non-uniform refinement thus producing a sharpened map of 3.37 Å. The local resolution was calculated in cryoSPARC. As for the paltusotine-bound SSTR2-Gi1 complex, a total of 1,302,282 particles were auto-picked and extracted in Relion 4.0-beta[51] with a box size of 256 pixels. 1,127,812 well-defined particles were selected after 2D classification to build the initial model. Following two rounds of 3D classifications, the best class determined by eye assessment was subjected to CTF refinement, Bayesian polishing[52], 3D auto-refine, post-process and produced a 3.24 Å sharpened map. The local resolution was estimated in RELION 4.0-beta. Details for each cryo-EM reconstruction could be found in Supplementary information, Table S1.

## Model building and structure refinement

The initial Gi1 and scFv16 models were created using the cryo-EM structure of the S1PR1–Gi1 complex (PDB: 7EW7), and the initial SSTR2 model was created by Swiss-model server[53]. UCSF Chimera1.14 was used to fit all models into the EM density map[54], followed by iterative rounds of manual adjustment and real-space refinement with Coot 0.8.9.2[55] and Phenix 1.19[56], respectively. MolProbity[57] was used to validate the models. The final structures had a satisfactory model geometry, and the comprehensive refining statistics could be found in Supplementary material, Table S1. UCSF ChimeraX1.1[58] and PyMOL were used to generate the structural figures. (https://pymol.org/2/).

## Bystander BRET (bBRET)

The human full-length β-arrestin 1 was cloned into pcDNA3.1+ vector with the Nluc of the BRET system and a 15-amino acid flexible linker (GGSGGGGSGGSSSGG) at the N-terminus. FYVE, the early endosomal marker[35] was cloned into pcDNA3.1+ vector with a 15-amino acid flexible linker (GGSGGGGSGGSSSGG) and C-terminally fused to a mVenus. Wild-type or mutant SSTR2, β-arrestin 1-Nluc and FYVE-mVenus were co-transfected in HEK293 cells at a ratio of 1:0.5:4 on six-well plates. The next day, the transfected cells were plated on white 96-well and incubated overnight, and then cells were incubated with 90 µL HBSS containing 20 mM HEPES and coelenterazine h (5 µM) at room temperature for 40 min. After incubation, ligand was added and the signal was measured using the Synergy H1 microplate reader (BioTek) to determine a BRET ratio.

## Statistical analysis

All data in this study were analyzed using GraphPad Prism 7 and were expressed as the error bars and the standard error of means ± SEM from at least $n = 3$ biologically independent experiments performed in triplicate. All curve data were normalized to the maximal wild-type and processed using a sigmoidal dose response curve in GraphPad Prism 7. Statistical differences between conditions were determined by one-way of variance ANOVA with Dunnett's test or T-test. The bias factors (β value) were determined as described in the previous reports[59] as the following equation:

$$\beta\,value = log\left(\left[\frac{E_{max}, P1}{EC_{50}, P1} \times \frac{EC_{50}, P2}{E_{max}, P2}\right]_{ligand} \times \left[\frac{E_{max}, P2}{EC_{50}, P2}\frac{EC_{50}, P1}{E_{max}, P1}\right]_{reference}\right)$$

(1)

where P1 is SSTRs Gi-mediated cAMP inhibition assay; P2, NanoBiT β-arrestin recruitments; β value > 0 denotes Gi protein biased, while β value < 0 indicates β-arrestin biased.

Parameters used in this equation were based on the curve fits of the combined datasets described above.

## Reporting summary

Further information on research design is available in the Nature Portfolio Reporting Summary linked to this article.

## Data availability

The cryo-EM density maps and structural data of SSTR2/octreotide (EMDB ID: EMD-33710; PDB ID: 7YAE) and SSTR2/paltusotine (EMDB ID: EMD-33708; PDB ID: 7YAC) are deposited in EMDB and PDB. The PDB codes and hyperlinks used in our structural alignment were listed below: AT1R-TRV023 (PDB code: 6OS1); µOR-Bu72 (PDB code: 5C1M) or FH210 (PDB code: 7SCG); the inactive SSTR2 (PDB code: 7UL5, colored in wheat); SSTR2-SST14 (PDB code: 7T10). Other data are available in the main text or the supplementary materials. Source data underlying Figs. 1–6 and Supplementary Figs. 5–9 are provided with this paper. Source data are provided with this paper.

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

## Acknowledgements

This research was supported by the National Natural Science Foundation of China (32100988 to W.Y.; 32200986 to J.Z.; 31972916 to Z.S.), Science and Technology Department of Sichuan Province (2021ZYD0080 to W.Y.), National Key R&D Program of China (2019YFA0508800 to Z.S.), the National Science Foundation for Excellent Young Scholars (32122052 to X.W.), National Natural Science Foundation Regional Innovation and Development (No. U19A2003 to X.W.), the China Postdoctoral Science Foundation (BX20220219 to Z.X.; 2022M712275 to J.Z.) and Post-Doctor Research Project, West China Hospital, Sichuan University (2021HXBH064 to J.Z.). We would like to thank the staff of Sichuan University West China Cryo-EM Center and Cryo-EM Center in Southern University of Science and Technology for cryo-EM data collection and the staff of the National Center for Protein Science in Shanghai for technical support. We thank the Sichuan University State Key Laboratory of Biotherapy Microscope platform for confocal assay. This work used resources from the Duyu High Performance Computing Center, Sichuan University, and Big Data Platform at West China Hospital of Sichuan University (WCH-BDP).

## Author contributions

H.F. designed the expression constructs and purified the SSTR2–Gi complex under the supervision of L.C.; J.Y., W.H., and J.Q. performed the cell signaling assays with the assistance of C.W., H.F., and S.S.; W.H. performed Internalization confocal assay. H.F. prepared the final samples for cryo-EM data collection with the assistance of C.Z., X.T., and Z.X.; C.Z. and X.T. prepared the cryo-EM grids, collected cryo-EM images, and performed map calculations under the supervision of Z.S.; C.Z., J.Z., and X.T. built and refined the structure models under the supervision of J.Z.; J.Z. analyzed the results and wrote the manuscript. R.C., W.Y., X.W., and Z.S. supervised the overall project and revised the manuscript.

## Competing interests

The authors declare no competing interests.
