## [Peer Review File · Nature Communications]

Prospect of acromegaly therapy: molecular mechanism of clinical drugs octreotide and paltusotineReviewers' Comments:

Reviewer #1:

Remarks to the Author:

In this manuscript, the authors reported molecular mechanism of octreotide and paltusotine for SSTR2. Additionally, the structural and mutagenesis assay provided some insight to the subtype-selectivity and biased signaling mechanism for SSTR2. Overall, I think the structures and functional analysis have some values to the SSTR2 pharmacology. However, the broad impact to the GPCR research field in regards to the subtype selectivity and biased signaling mechanism are lacking.

Major comments:

1. So far there have been at least five active-state and one inactive-state SSTR2 structures/papers published (1.Skiniotis et al, NSMB, 2022 Feb; 2.Lee et al, eLife, 2022 Apr; 3.Mao et al, Cell Research, 2022 May; 4.Tian et al, Cell Discovery, 2022 May; 5.Zhao et al, Cell Research, 2022 June). Among these papers, some ligands are commonly used, such as octreotide.

(a) what's the NEW insight that this current study can provide to the GPCR field?

(b) how could this current study ADVANCE the SSTR2 research compared to existing studies?

(c) while I appreciate the independent work of this manuscript, a through comparison between different ligand binding modes, and the comparison of the same ligand-bound structures (to validate structural reliability by independent methods) would be needed.

2. Seen from the Supplementary Fig. 5a, the octreotide also showed higher affinity and efficacy on SSTR2 than other subtypes of SSTR family members, is there any explanation on it based on the structure?

3. For the selectivity analysis, some comparison between paltusotine and the small molecule L-054,264 might be helpful.

4. For the signal bias part, the relationship between microswitch residues and signal bias with different ligands were not clarified. Is the response to signal bias ligand dependent?

5. What's the influence of the microswitch residues on the receptor internalization for both ligands?

6. Since the authors repetitively emphasized the subtype selectivity and biased signaling, I'm not clear on what pharmacological profile is therapeutically beneficial for SSTR2 drug discovery? (subtype selectivity? Signaling bias?) is there any clinical evidence?

7. Some structural analysis does not seem reliable, for example, Line 218-219: "Superposition of the active state of SSTR2 with SSTR3 (predicted active model from GPCRdb)..." this is a comparison between an experimental structure for protein A and predicted model for protein B, then why we need experimental structure?

8. For many side chain analysis, it is in the context of a $\sim 3.3\text{\AA}$ resolution cryo-EM map, and the side chain placement, at least for some of them, could be ambiguous. Therefore, for some key residues (engaging core interactions with the two ligands) on the receptor, density map should be shown.

Minor comments:

1. The "Discussion" part looks more like a "Summary", which should be further revised. Especially, the authors suggested future studies on inactive SSTR2 structure which has been recently reported by Skiniotis, G. et al (PDB 7UL5).

2. There are some typos and grammatical errors which need careful check. For example, Line 215, by "combing with ..." should be combining.

Reviewer #2:

Remarks to the Author:

In this manuscript, Zhao, J. et al report the cryo-EM structures of the human somatostatin receptor 2 (SSTR2) - Gi complexes bound with either the first-generation peptide drug octreotide or the new-generation small molecule drug paltusotine. The structures reveal the molecular mechanism of the subtype selectivity of paltusotine towards the SSTR2 over other SSTR subtypes. What is more

interesting, paltusotine exhibits better G-protein biased signaling compared to octreotide and exhibits a potential better performance in promoting apoptosis of pituitary tumor cells.

Due to the importance of SSTR2 as drug target, many groups have solved the structures of SSTR2 bound with different ligands. I found at least five publications through a quick search:

1 Zhao, W. et al. Structural insights into ligand recognition and selectivity of somatostatin receptors. *Cell Res* 32, 761-772, doi:10.1038/s41422-022-00679-x (2022).

2 Robertson, M. J., Meyerowitz, J. G., Panova, O., Borrelli, K. & Skiniotis, G. Plasticity in ligand recognition at somatostatin receptors. *Nat Struct Mol Biol* 29, 210-217, doi:10.1038/s41594-022-00727-5 (2022).

3 Heo, Y. et al. Cryo-EM structure of the human somatostatin receptor 2 complex with its agonist somatostatin delineates the ligand-binding specificity. *Elife* 11, doi:10.7554/eLife.76823 (2022).

4 Chen, L. N. et al. Structures of the endogenous peptide- and selective non-peptide agonist-bound SSTR2 signaling complexes. *Cell Res* 32, 785-788, doi:10.1038/s41422-022-00669-z (2022).

5 Bo, Q. et al. Structural insights into the activation of somatostatin receptor 2 by cyclic SST analogues. *Cell Discov* 8, 47, doi:10.1038/s41421-022-00405-2 (2022).

However, the authors only cited two of them in this manuscript. I think all of these five papers should be cited and discussed. For example the NMSB paper also reports the SSTR2-octreotide structure, but with a higher resolution (2.7Å vs 3.37Å). The readers would be interested to know if the structure models are similar or different in these two works. Furthermore, the authors claim that "Future studies would be highly informative to solve the inactive SSTR2 and" (line 333) while the inactive SSTR2 structure has been solved and reported in the Cell research paper by Zhao, W. et al.

The highlight of this work, in my mind, is the studies of β -arrestin recruitment and internalization. However, I am not totally convinced that N2766.55 and F2947.35 play key roles to the signal bias of SSTR2 when sensing octreotide. The N2766.55A and F2947.35A mutation indeed cause larger reduction in octreotide stimulated β -arrestin recruitment compared to Gi signaling, but I am not sure if the differences are due to biased signaling or just due to impaired affinity. For example, N2766.55A causes ~90 fold increase in EC50 for octreotide stimulated Gi activation and ~540 fold increase for β -arrestin recruitment. It's possible that this mutation just reduced the affinity of octreotide and made this ligand less functional in activating SSTR2. This affinity reduction has a larger effect on β -arrestin recruitment assay because β -arrestin generally binds weaker to the receptor than G protein. I also noticed that the current β -arrestin recruitment assays were done without co-transfection of GRK. It would be more convincing if the authors could repeat the assay in presence of GRK and check if these two mutations indeed abolish β -arrestin recruitment.

One interesting question is why paltusotine is more G protein biased compared to octreotide, as this information may guide future drug design. But the authors did not really address this question.

Other minor suggestions include:

1. In line 29, the authors claim that "drug resistance occurs in a subset of patients, which may be correlated with SSTR subtype selectivity or cell surface expression." I understand the correlation between 'drug resistance' and 'cell surface expression'. But I don't quite understand why 'subtype selectivity' correlates with 'drug resistance'.

2. The confocal fluorescence microscopy images are confusing to me. For example, in Fig.6c and 6d, it looks like paltusotine induces as much internalization as octreotide does in WT SSTR2. While in Fig. 6c, it looks like octreotide induces as much internalization for N2766.55A and F2947.35A mutants compared to WT SSTR2. These results are not consistent with the main conclusion of the manuscript.

Point-to-point response letter

We thank the referees for their valuable time in reviewing our manuscript and the constructive suggestions that they have provided. Please find our responses to the specific comments raised by the reviewers below. We have copied each comment in *Italic*, which is followed by our own point-by-point response in **blue**, including details about the corresponding changes to the manuscript.

Reviewer #1:

In this manuscript, the authors reported molecular mechanism of octreotide and paltusotine for SSTR2. Additionally, the structural and mutagenesis assay provided some insight to the subtype-selectivity and biased signaling mechanism for SSTR2. Overall, I think the structures and functional analysis have some values to the SSTR2 pharmacology. However, the broad impact to the GPCR research field in regards to the subtype selectivity and biased signaling mechanism are lacking.

Response: Thank you so much for taking the time to evaluate our work. We appreciate your constructive comments that improved our study. In the revised manuscript, we have carried out additional experiments and included the results about the mechanisms of ligand selectivity and bias signal.

Major comments:

1. So far there have been at least five active-state and one inactive-state SSTR2 structures/papers published (1.Skiniotis et al, NSMB, 2022 Feb; 2.Lee et al, eLife, 2022 Apr; 3.Mao et al, Cell Research, 2022 May; 4.Tian et al, Cell Discovery, 2022 May; 5.Zhao et al, Cell Research, 2022 June). Among these papers, some ligands are commonly used, such as octreotide.

(a) what's the NEW insight that this current study can provide to the GPCR field?

(b) how could this current study ADVANCE the SSTR2 research compared to existing studies?

Response: We thank for the reviewer's insightful comments. Previous studies mentioned by reviewer described about the mechanism of ligand recognition, receptor activation, as well as subtype selectivity of the group 2 (SSTR2/3/5) receptors vs. group 1 (SSTR1/4). SSTR2 signals via activation of Gi protein and engages β -arrestin to mediate distinct cellular signaling events, however, pharmacological properties of different types of ligands (peptide octreotide and small molecule paltusotine) remain unclear. Here, one of the new insights in our work is that we first characterized the pharmacological profiles of octreotide and paltusotine at SSTR2. On the other hand,

our study determined the selective mechanism of paltusotine for SSTR2 among group 2 SSTRs (SSTR2/3/5). Similar with other GPCR subtypes, SSTRs distributed in different tissues and regulated divergent physiological processes. Therefore, designing selective ligands that can achieve receptor subtype selectivity and specific receptor signaling and even control on- or off-target side effects, could be safer therapeutic strategy in GPCR drug discovery field.

Compared with the published studies about SSTR2, we supplied the mechanisms of bias signaling and SSTRs group 2 subtype selectivity in this manuscript. In addition, by performing cell-based G-protein activation assay, β -arrestin recruitment assay, and receptor internalization analysis, we demonstrated pharmacological features of different generations of drugs targeting SSTR2. Furthermore, we measured the pituitary tumor cell GH3 apoptosis after administration of octreotide and paltusotine by using flow cytometry. In general, our finding provides comprehensive insights into understanding the structural basis of SSTR2 and the functional actions of divergent drugs.

(c) while I appreciate the independent work of this manuscript, a through comparison between different ligand binding modes, and the comparison of the same ligand-bound structures (to validate structural reliability by independent methods) would be needed.

Response: Thanks for the valuable comment. We do agree that the comparison of the structures solved in this study with those in the previous studies are significative and conclusive. In the revised manuscript, i) we first carried out structural comparison of peptide ligands binding modes in SSTR2, including agonist octreotide and endogenous agonist SST14. ii) The detailed ligand recognition and the critical microswitches required for receptor activation were further analyzed in the different states of the receptor. iii) In addition, we compared the binding mode of small molecules in SSTR2. The related descriptions are presented below and the related discussion has been included in the result section 2 “Recognition mechanism of octreotide by SSTR2” (lines 124-128, 134-135, 140-143) and the discussion section.

Table R1. The up-to-date structures of SSTR2.

PDB ID	Resolution (Å)	Ligand	State	Type of G α protein	reference
7T10	2.50	SST14	Active	Gi3	Nat Struct Mol Biol
7T11	2.70	Octreotide	Active	Gi3	Nat Struct Mol Biol
7UL5	3.10	—	Inactive	—	Skiniotis To be published
7WIC	2.80	SST14	Active	Gi1	Cell Res
7WIG	2.70	L-054,264	Active	Gi1	Cell Res

7WJ5	3.72	SST14	Active	Gi1	Elife
7XAT	2.85	SST14	Active	Gi1	Cell Discov.
7XAU	2.97	Octreotide	Active	Gi1	Cell Discov.
7XAV	2.87	Lanreotide	Active	Gi1	Cell Discov.
7XMR	3.10	SST14	Active	Gi1	Cell Res
7XN9*	2.60	L-054,522	Intermediate	—	Cell Res
7XNA*	2.65	CYN 154806	Inactive	—	Cell Res
7YAE	3.37	Octreotide	Active	Gi1	In this study
7YAC	3.24	Paltusotine	Active	Gi1	In this study

* Structures determined by the X-ray diffraction strategy.

As is shown in Table R1, we summarized the reported structures of SSTR2 so far. The structures of octreotide-bound SSTR2 have been determined by *Skiniotis* and *Tian* groups (PDB ID: 7T11 and 7XAU), structural comparisons of SSTR2-Gi in complex with octreotide with the previous two signaling complexes reveal the nearly identical assembly architecture (Fig. R1a-d), with a RMSD of 0.82 – 1.07 Å for the C α atoms of the receptor. Additionally, the binding of SSTR2 with octreotide in these three complex structures exhibit the same recognition mechanism, despite different Gi proteins (Gi1 or Gi3) couple.

Both approved drugs octreotide and lanreotide contain pharmacological core region ((D)-Tyr4 and Lys5) that is necessary for receptor activation and inserts into the bottom of the orthosteric pocket, subsequently triggering the rearrangement of microswitches for receptor activation (Fig. R1e, f), for instance, the toggle switch W^{6,48}, PIF and DRY motifs, as well as NPxxY motif. These key residues or motifs in the structures of octreotide- or lanreotide-bound SSTR2 exhibit similar conformation as endogenous peptide SST14-bound structure.

Moreover, two structures of small-molecules L-054,522- and L-054,264-bound SSTR2 were reported by *Mao* and *Zhao* groups, respectively. It is noteworthy that paltusotine displays different scaffolds from L-054,522 and L-054,264. Structural comparison of paltusotine-SSTR2 structure with these two structures reveal a common binding site as well as extended binding sites for specific moieties (Fig. R1g). All of the three molecules occupy the core binding region, L-054,264 and L-054,522 share the extended binding region, whereas paltusotine occupies a minor pocket. In detail, the conformation of the residues involved in the core binding region of the three SSTR2 structures are almost identical (D122^{3,32}, Q126^{3,36}, F208^{5,38} and Y302^{7,43}) (Fig. R1h). The large 3,5-difluorophenyl moiety of paltusotine is placed toward the extracellular end of TM2, creating the minor pocket formed by Y50^{1,39}, L99^{2,60}, Q102^{2,63}, V103^{2,64} and D295^{7,36} in SSTR2 (Fig. R1h). While L-054,264 places into the extended binding

region formed by TM6 and TM7, and forms extensive interactions with F275^{6.54}, N276^{6.55}, S279^{6.58}, L290^{7.31} and F294^{7.35}, which are absent in paltusotine binding (Fig. R1h). As for L-054,522, it is much larger than paltusotine and L-054,264 in size. It also interacts with the paltusotine binding minor pocket (L99^{2.60}, Q102^{2.63}, V103^{2.64} and D295^{7.36}), on the other hand, L-054,522 extends into TM6, TM7 and ECL3 like L-054,264, and forms extensive interactions with F275^{6.54}, N276^{6.55}, L290^{7.31} and P286^{ECL3} (Fig. R1i). L-054,522 is just like the chimera of L-054,264 and paltusotine.

Fig. R1: Structural comparison of SSTR2 bound to different types of ligands.

a-d: Structural comparison of SSTR2-octreotide structures determined by previous studies with that

in this study. **a.** Overall structural comparison of SSTR2-octreotide in our study with SSTR2^{7T11}. SSTR2 in our study is shown as cartoon and colored in green-cyan, octreotide is shown as sticks and colored in light pink; SSTR2 in SSTR2^{7T11} is shown as cartoon and colored in gray, octreotide is shown as sticks and colored in yellow. **b.** The binding poses of octreotide in the two structures. **c.** Structural comparison of the residues in the octreotide binding pocket of SSTR2 in the two structures. Key residues in SSTR2 are shown as sticks. **d.** Conformational comparison of the microswitch residues of SSTR2 in these two structures. Key residues are shown as sticks. (see also in supplementary Fig. 4b-c)

e: Structural alignment of SSTR2-octreotide structure solved by this study (colored in green-cyan) with the inactive SSTR2 structure (PDB code: 7UL5, colored in wheat). Microswitch residues are shown as sticks. (see also in supplementary Fig. 4a)

f: Structural alignment of SSTR2-octreotide structure solved by this study (colored in green-cyan) with the lanreotide-bound (PDB code: 7XAV, colored in gray) or SST14-bound SSTR2 structure (PDB code: 7T10, colored in wheat). Microswitch residues are shown as sticks.

g: Structural comparison of paltusotine (cyan), L-054,264^{7WIG} (wheat) and L-054,522^{7XN9} (yellow). Small molecules are shown as sticks and SSTR2 is shown in an electrostatic surface representation. (see also in supplementary Fig. 7b)

h: Structural comparison of SSTR2-paltusotine and SSTR2-L-054,264^{7WIG}. SSTR2 in our study is shown as cartoon and colored in violet, paltusotine is shown as sticks and colored in cyan. SSTR2 in SSTR2-L-054,264^{7WIG} is shown as cartoon and colored in lime green, L-054,264 is shown as sticks and colored in wheat. Key residues involved in ligand binding in SSTR2 are shown as sticks, residues crucial for both ligands are labeled in black; residues only crucial for paltusotine binding are labeled in violet; the residues only involved in L-054,264 binding are labeled in wheat.

i: Structural comparison of SSTR2-paltusotine and SSTR2-L-054,522^{7XN9}. SSTR2 in our study is shown as cartoon and colored in violet, paltusotine is shown as sticks and colored in cyan. SSTR2 in SSTR2-L-054,522^{7XN9} is shown as cartoon and colored in wheat, L-054,522 is shown as sticks and colored in yellow. Key residues involved in ligand binding in SSTR2 are shown as sticks, residues crucial for both ligands are labeled in black; residues only crucial for paltusotine binding are labeled in violet; residues only involved in L-054,522 binding are labeled in yellow.

2. Seen from the Supplementary Fig. 5a, the octreotide also showed higher affinity and efficacy on SSTR2 than other subtypes of SSTR family members, is there any explanation on it based on the structure?

Response: Thank you for the question. Indeed, our data reveal that octreotide showed higher efficacy on SSTR2 (SSTR2 > SSTR3/5 > SSTR1/4), which is consistent with the previous study¹. In the revised manuscript, we analyzed the possible reason why octreotide exhibited higher efficacy than other subtypes of SSTR members, and then highlighted some points based on the structural comparison as well as mutagenesis studies.

First of all, combined with the previously published papers of SSTR2^{2,3} as well as

sequence alignment of SSTRs, the residues F^{7.35}, N^{6.55}, and Q^{2.63} were found to participate in subtype selectivity of octreotide on SSTR2 over group 1 SSTRs (SSTR1/4), which were conserved among SSTR2, R3 and R5. Accordingly, mutating these three residues in SSTR2 to the corresponding residues in SSTR1 (F^{7.35}S and N^{6.55}Q, Q^{2.63}S) markedly reduced the octreotide binding efficacy of SSTR2. Consistently, as was shown in our structure, F294^{7.35}S substitution probably abolish the hydrophobic interactions, whereas N276^{6.55} that is close to Trp4 in octreotide and N276^{6.55}Q replacement could result in steric clash with octreotide (Fig. R2a). In addition, Q102^{2.63} is observed to make direct interaction with Thr6 in octreotide-bound SSTR2 structure (Fig. R2a), while the equivalent residue S^{2.63} in SSTR1 and R4 could disrupt such contact and weaken the activation efficacy of octreotide. In a word, these residues contributing to the stable binding of octreotide with group 2 SSTRs (SSTR2/3/5) over group 1 SSTRs (SSTR1/4).

Secondly, octreotide showed moderate affinity for SSTR3/5 over SSTR2. To clarify the subtype selectivity of octreotide within group 2 SSTRs (SSTR2/3/5), we combined the sequence alignment with structural analysis (Fig. R2a, b) and focused on the non-conserved residues within the octreotide binding pocket. As was shown in our structure, T194^{45.51} interacts with Thr6 in octreotide, and it was replaced by histidine or asparagine in SSTR3 or R5, which might be responsible for the reduced octreotide binding affinity of SSTR3/5. Our cAMP inhibition assays showed that T194^{45.51}H substitution in SSTR2 (Fig. R2c), which may impair the contact between octreotide and SSTR2, significantly reduced the receptor activation.

In the revised manuscript, we have included related description in lines 375-378.

Fig. R2: Selectivity of octreotide among SSTRs family.

a: Octreotide binding pocket in SSTR2.

b: Sequence alignment of the residues within the octreotide binding pocket.

c: cAMP inhibition assays of T194^{45.51}H SSTR2. (See also in Supplementary Fig. 7j)

3. For the selectivity analysis, some comparison between paltusotine and the small molecule L-054,264 might be helpful.

Response: Thanks for the helpful suggestion. We first compared the binding mode of

paltusotine and L-054,264 in SSTR2, the common binding pocket formed by the key residues D122^{3,32}, Q126^{3,36}, F208^{5,38} and Y302^{7,43} in SSTR2 was observed in both structures. Notably, the indene-piperidine moiety of L-054,264 extends toward TM6 and TM7, forming extensive interactions with F275^{6,54}, N276^{6,55}, S279^{6,58}, L290^{7,31} and F294^{7,35}; whereas the large 3,5-difluorophenyl moiety of paltusotine occupies a minor pocket constituted by TM1, TM2 and TM7, making direct contacts with Y50^{1,39}, L99^{2,60}, Q102^{2,63}, V103^{2,64} and D295^{7,36} (Fig. R3a).

Consistent with previous reporting⁴, our results of cAMP inhibition measurement revealed that both L-054,264 and paltusotine behave more potent selectivity for SSTR2 relative to other subtypes (Fig. R3b, c). In the study published by *Mao* group discussed the selective mechanism of L-054,264 on SSTR2, and they indicated that the residues F275^{6,54}, F294^{7,35} and N276^{6,55} determined the selective recognition of L-054,264 by SSTR2⁵, however, paltusotine lacks the direct contacts with these residues due to different binding pose in SSTR2. Furthermore, we tended to focus on the minor binding site engaged in paltusotine binding. We next generated substitution of the non-conserved residues from sequence alignment (Fig. R3d), and the results of cAMP inhibition measurement revealed that replacement of V103^{2,64} and T194^{45,51} with the corresponding residues (N103^{2,64} and H194^{45,51}) in SSTR3 reduced the activation of SSTR2 induced by paltusotine (Fig. R3e), meanwhile, both N101^{2,64}V and H192^{45,51}T substitutions in SSTR3 increased the receptor activation (Fig. R3f). Thus, our results indicated that V103^{2,64} and T194^{45,51} involved in SSTR2 selectivity when sensing paltusotine. These findings indicated the multifaceted subtype selective mechanism. The related descriptions have been included in the result section 4 “Selectivity of paltusotine for SSTR subtypes”, lines 223-235.

Fig. R3: The recognition mechanism of paltusotine by SSTR2.

a: Structural comparison of SSTR2-paltusotine and SSTR2-L-054,264^{7WIG}. SSTR2 in our study is shown as cartoon and colored in violet, paltusotine is shown as sticks and colored in cyan. SSTR2 in SSTR2-L-054,264^{7WIG} is shown as cartoon and colored in lime green, L-054,264 is shown as sticks and colored in wheat. Key residues involved in ligand binding in SSTR2 are shown as sticks, residues only crucial for paltusotine binding are labeled in violet; residues only involved in L-054,264 binding are labeled in wheat. (see also in supplementary Fig. 7c)

b: Gi signaling measurement of SSTR1–5 activated by L-054,264 using cAMP inhibition assays. Data represent mean ± SEM from three independent experiments for SSTR1-5.

c: Gi signaling measurement of SSTR1–5 activated by paltusotine using cAMP inhibition assays. Data represent mean ± SEM from three independent experiments for SSTR1-5. (see also in Fig. 4a)

d: Sequence alignment of the residues engaged in paltusotine binding in SSTR2 with other SSTR subtypes. (see also in Fig. 4b)

e, f: The effects of the residue substitution of N103^{2.64}V or T194^{45.51}H in SSTR2 (**e**) and V101^{2.64}N or H192^{45.51}T in SSTR3 (**f**) on the paltusotine-induced cAMP inhibition. Data represent mean ± SEM from three independent experiments. (see also in Fig. 4c, d)

4. For the signal bias part, the relationship between microswitch residues and signal bias with different ligands were not clarified. Is the response to signal bias ligand dependent?

Response: Thanks for the valuable comment. According to the suggestion, we further investigated the contribution of the key microswitches of SSTR2 to the signal bias property, including toggle switch W^{6.48}, PIF, DRY and NPxxY motifs. The result of our cAMP inhibition and β-arrestin recruitment assays indicated that alanine substitution of key residues from those microswitches significantly reduced the potency of β-arrestin recruitment, but only slightly influenced the G-protein activation induced by octreotide (Fig. R4a-f). Particularly, in agreement with octreotide stimuli, those mutants of SSTR2 exhibited similar manner in G-protein activation or β-arrestin recruitment in response to paltusotine (Fig. R4g-l). Collectively, our finding demonstrated that critical microswitches contribute to the β-arrestin signal bias of SSTR2, the relationship between microswitch residues with signal bias of GPCR have been investigated in previous studies⁶⁻⁸. Meanwhile, the bias property of SSTR2 microswitch residues were similar when sensing both octreotide and paltusotine, which indicates that the microswitch residues involved signal bias property is not ligand dependent. The related descriptions have been included in the result section 5 “Signal bias properties of SSTR2 with different ligands”, lines 310-314.

Fig. R4: Bias profiles of microswitch residues in SSTR2 induced by octreotide and paltusotine.

a-f: The effects of alanine substitution of microswitch residues W^{6.48} in C^{6.47}W^{6.48}xP^{6.50}, Y^{7.53} in N^{7.49}P^{7.50}xxY^{7.53}, R^{3.50} in D^{3.49}R^{3.40}Y^{3.51} (a-c) and P^{5.50}I^{3.40}F^{6.44} (d-f) and β-arrestin recruitment (b, e) induced by octreotide. Bias factors of these mutants relative to wild-type induced by octreotide (c, f). (Fig. R4c see also in Supplementary Fig. 8e)

g-l: The effects of alanine substitution of microswitch residues (W^{6.48} in C^{6.47}W^{6.48}xP^{6.50}, Y^{7.53} in N^{7.49}P^{7.50}xxY^{7.53}, and R^{3.50} in D^{3.49}R^{3.40}Y^{3.51}) in SSTR2 on cAMP inhibition (g, j) and β-arrestin recruitment (h, k) induced by octreotide. Bias factors of these mutants relative to wild-type induced by paltusotine (i, l). (Fig. R4i see also in Supplementary Fig. 8e)

5. What's the influence of the microswitch residues on the receptor internalization for both ligands?

Response: We thank the reviewer for pointing out detecting the influence of the microswitch residues on the receptor internalization. According to the suggestion, we explored the impact of the microswitch residues on SSTR2 internalization by performing ELISA assay and receptor trafficking by Bystander BRET assays, with co-transfecting with FYVE, an early endosome sensor^{9,10}. Consistent with our β -arrestin recruitment assays, alanine substitution of microswitch residues resulted in reduced receptor internalization when sensing both octreotide (Fig. R5a, b) and paltusotine (Fig. R5c, d). Therefore, our data suggests that the internalization of SSTR2 can be affected by microswitch residues. The related descriptions have been included in the result section 6 “Influence of SSTR2 internalization by the ligand-induced β -arrestin signal”, lines 324-331.

Fig. R5: Impact of microswitch residues on the internalization of SSTR2 induced by octreotide and paltusotine. (see also in Supplementary Fig. 10)

a-b: The effects of alanine substitution of microswitch residues W^{6.48} in C^{6.47}W^{6.48}xP^{6.50}, Y^{7.53} in N^{7.49}P^{7.50}xxY^{7.53}, R^{3.50} in D^{3.49}R^{3.50}Y^{3.51} (a) and P^{5.50}I^{3.40}F^{6.44} (b) on the internalization of SSTR2 induced by octreotide detected by Bystander BRET based trafficking assays in the presence of the early endosome marker FYVE. Data represent mean \pm SEM from three independent experiments.

c-d: The effects of alanine substitution of microswitch residues W^{6.48} in C^{6.47}W^{6.48}xP^{6.50}, Y^{7.53} in N^{7.49}P^{7.50}xxY^{7.53}, R^{3.50} in D^{3.49}R^{3.50}Y^{3.51} (c) and P^{5.50}I^{3.40}F^{6.44} (d) on the internalization of SSTR2 induced by paltusotine detected by Bystander BRET based trafficking assays in the presence of the early endosome marker FYVE. Data represent mean \pm SEM from three independent experiments.

6. Since the authors repetitively emphasized the subtype selectivity and biased signaling, I'm not clear on what pharmacological profile is therapeutically beneficial for SSTR2 drug discovery? (subtype selectivity? Signaling bias?) is there any clinical evidence?

Response: We gratefully appreciate for your valuable comments. SSTR2 is a therapeutic target for neuroendocrine tumors treatment. Approved drugs and next generation agents with subtype selectivity have always been pursued in drug discovery targeting SSTR2^{11,12}. Achieving SSTRs subtype selectivity could facilitate to improve the efficacy of potential drugs and avoid side effects caused by off-target¹³. In addition, previous studies have demonstrated that β -arrestin could drive desensitization and internalization of SSTR2 induced by SST analogs¹⁴. During the administration of SST analog drug octreotide for acromegalic patients, the expression level of SSTR2 occurred down-regulated¹⁵. Further, an experimental study from over 30 acromegalic patients revealed that the expression level of β -arrestin 1 had an crucial role in the modulation of SST analogs drug efficacy as well as growth hormone secretion, which suggests that lower expression of β -arrestin 1 in pituitary adenomas may be associated with decreased recycling rate of SSTR2 and better SST analog response^{16,17}. It is noteworthy that a recent reporting from Crinetics Pharmaceuticals, Inc. and University of Texas Health Science center suggested that internalization might be thought to limit the therapeutic effect of SST analogs, and they announced a small molecule named paltusotine (under clinical phase 3 currently) with improved efficacy by activating Gi-biased signaling and reducing desensitization and internalization of SSTR2¹⁸. Collectively, together with previous literatures, we provide insights to understand the correlation of subtype selectivity and signaling bias of SSTR2 with therapeutic efficacy. The related descriptions have been included in the “introduction” section paragraph 2 and 3, lines 63-65 and 74-79.

7. Some structural analysis does not seem reliable, for example, Line 218-219: “Superposition of the active state of SSTR2 with SSTR3 (predicted active model from GPCRdb)” this is a comparison between an experimental structure for protein A and predicted model for protein B, then why we need experimental structure?

Response: Thanks for your helpful suggestion. We apologize about the confusing description in the manuscript. In the revised manuscript, we aim to examine the subtype selectivity of paltusotine for distinguishing group 2 SSTRs (SSTR2, R3 and R5), so the accurate three-dimensional structure of SSTR2 determined by experimental technique is being pursued. Structural visualization of paltusotine-bound SSTR2 complex can confirm the interaction mode of ligand in the orthosteric site of SSTR2, and sequence alignment among SSTRs could provide an opportunity for us to discuss the differences. Furthermore, by carefully analyzing the binding model of paltusotine and octreotide with SSTR2, we eventually found 6 residues that were not conserved among group 2 SSTRs with the ligand binding pocket. We next examined the role of divergent residues in receptor activation by performing residue substitution and measuring cAMP inhibition efficacy, hopefully, the predicted model of SSTR3 based on homologous

structure could aid in understanding the results of cell-based assays as well as the mechanism of subtype selectivity of paltusotine for SSTR2 among group 2 SSTRs. To avoid this confusion, we moved the related main figure to supplementary files in the revised manuscript.

8. For many side chain analysis, it is in the context of a $\sim 3.3\text{\AA}$ resolution cryo-EM map, and the side chain placement, at least for some of them, could be ambiguous. Therefore, for some key residues (engaging core interactions with the two ligands) on the receptor, density map should be shown.

Response: Thanks for the reviewer's helpful suggestion. In the revised manuscript, we have shown the density maps of the side chain of key residues engaged in ligand recognition, further supporting the believable modes of ligand-bound SSTR2 (Fig. R6). In addition, we included the figure of the density maps in the Supplementary Fig. 3.

Fig. R6: The electron density map of the key residues involved in ligand binding in our two structures. (see also in Supplementary Fig. 3)

a-c: The electron density map of the residues in the octreotide binding pocket. SSTR2 is shown in a cartoon diagram and colored in green-cyan; octreotide is shown as sticks and colored in light pink; density map is shown in gray meshes at contour level of 0.4.

d-f: The electron density map of the residues in the paltusotine binding pocket. SSTR2 is shown in a cartoon diagram and colored in violet; paltusotine is shown as sticks and colored in cyan; density map is shown in gray meshes at contour level of 0.004.

Minor comments:

1. The "Discussion" part looks more like a "Summary", which should be further revised. Especially, the authors suggested future studies on inactive SSTR2 structure which has been recently reported by Skiniotis, G. et al (PDB 7UL5).

Response: Thanks for the constructive suggestion. In the revised manuscript, we have included structural comparison of active with inactive state, receptor activation, and subtype selectivity as well as biased therapeutic of SSTR in the revised discussion section. Our findings provide insights into understanding the safe window of therapeutic agents with biased pharmacology in GPCR drug discovery field.

We first checked the inactive structure of SSTR2 without ligand binding (PDB code: 7UL5) and the antagonist peptide CYN 154806-bound inactive structure of SSTR2 (PDB code: 7XNA). As suggested, the structural comparison of our structures with these two inactive structures reveals that the agonist paltusotine inserts in the core binding region of SSTR2 deeply, forming direct interaction with the side chain of Q126^{3,36}, simultaneously, the conformational displacement of Q126^{3,36} may cause rearrangement of the microswitches such as W269^{6,48}, I130^{3,40} and F265^{6,44} in PIF motif, finally altering the conformation of TM5 and TM6 to achieve G-protein coupling (Fig. R7).

Similar with other GPCR subtypes, SSTRs are distributed in different tissues, regulating divergent physiological processes. SSTR2 is a valuable drug target for the treatment of many diseases such as acromegaly. The high sequence homology among SSTR subtypes and divergent bias profiles of SSTR2 call for the development of therapeutic drugs toward specific subtypes and signaling pathways. Here, we characterized the pharmacological profiles of the clinical drug paltusotine and the first-generation drug octreotide. Structural determination of paltusotine-bound and octreotide-bound SSTR2-Gi signaling complexes elucidate the molecular mechanism of the recognition of paltusotine and octreotide by SSTR2. During preparation of our manuscript, several individual groups reported the structures of SSTR2 bound to octreotide and other different types of ligands^{2,3,5,19,20}. We noticed that the octreotide-bound structures from three independent groups exhibit nearly identical conformation. In addition, these studies profiled the subtype selectivity of ligand to group 2 SSTRs (SSTR2/3/5) over group 1 SSTRs (SSTR1/4). Herein, we further investigated the mechanism that paltusotine discriminates specific subtype from group 2 SSTRs via an unusual minor pocket in SSTR2. Emerging of the structures of other SSTRs will be helpful to understand selective mechanism and could provide more information to design selective small molecules toward specific SSTR subtype.

Previous clinical data showed that the expression level of SSTR2 from acromegalic patients occurred down-regulated¹⁵ during the administration of SST analog drug octreotide, and the expression level of β -arrestin 1 in over 30 acromegalic patients were demonstrated to be important for modulation of the efficacy of SST analog drugs as well as growth hormone secretion, which suggests that lower expression of β -arrestin 1 in pituitary adenomas was associated with decreasing recycling rate of SSTR2 and better SST analog response^{16,17}. The result of our

functional assay revealed that paltusotine displayed more G protein-biased property compared with octreotide. By further inspecting the differences between octreotide- and paltusotine-induced SSTR2 biased signaling, we found that paltusotine loses the interactions with I284^{ECL3}, K291^{7.32}, N276^{6.55} and F294^{7.35}, which participated in the recruitment of β -arrestin induced by octreotide. Our study, to a certain extent, contributes to the understanding of the functional bias of ligands and guiding rational drug design targeting SSTRs. Thus, designing selective ligands that can achieve receptor subtype selectivity or specific receptor signaling and even control on- or off-target side effects, could be safer therapeutic agents in GPCR drug discovery field.

Fig. R7: Structural comparison of SSTR2-paltusotine structure determined in our study with the inactive structures reported previously (7UL5 and 7XNA).

2. There are some typos and grammatical errors which need careful check. For example, Line 215, by “combing with ...” should be combining.

Response: We thank the reviewer for pointing out this issue. We have checked through the whole text carefully and corrected some typos and grammatical errors in the revised manuscript.

Reviewer #2:

In this manuscript, Zhao, J. et al report the cryo-EM structures of the human somatostatin receptor 2 (SSTR2) - Gi complexes bound with either the first-generation peptide drug octreotide or the new-generation small molecule drug paltusotine. The structures reveal the molecular mechanism of the subtype selectivity of paltusotine towards the SSTR2 over other SSTR subtypes. What is more interesting, paltusotine exhibits better G-protein biased signaling compared to octreotide and exhibits a potential better performance in promoting apoptosis of pituitary tumor cells.

Due to the importance of SSTR2 as drug target, many groups have solved the structures of SSTR2 bound with different ligands. I found at least five publications through a quick search:

1 Zhao, W. et al. Structural insights into ligand recognition and selectivity of somatostatin receptors. *Cell Res* 32, 761-772, doi:10.1038/s41422-022-00679-x (2022).

2 Robertson, M. J., Meyerowitz, J. G., Panova, O., Borrelli, K. & Skiniotis, G. Plasticity in ligand recognition at somatostatin receptors. *Nat Struct Mol Biol* 29, 210-217, doi:10.1038/s41594-022-00727-5 (2022).

3 Heo, Y. et al. Cryo-EM structure of the human somatostatin receptor 2 complex with its agonist somatostatin delineates the ligand-binding specificity. *Elife* 11, doi:10.7554/eLife.76823 (2022).

4 Chen, L. N. et al. Structures of the endogenous peptide- and selective non-peptide agonist-bound SSTR2 signaling complexes. *Cell Res* 32, 785-788, doi:10.1038/s41422-022-00669-z (2022).

5 Bo, Q. et al. Structural insights into the activation of somatostatin receptor 2 by cyclic SST analogues. *Cell Discov* 8, 47, doi:10.1038/s41421-022-00405-2 (2022).

However, the authors only cited two of them in this manuscript. I think all of these five papers should be cited and discussed. For example, the NMSB paper also reports the SSTR2-octreotide structure, but with a higher resolution (2.7Å vs 3.37Å). The readers would be interested to know if the structure models are similar or different in these two works. Furthermore, the authors claim that “Future studies would be highly informative to solve the inactive SSTR2 and” (line 333) while the inactive SSTR2 structure has been solved and reported in the Cell research paper by Zhao, W. et al.

Response: We thank the reviewer for taking the time to evaluate our work. We do agree that it is of importance to compare available structures of SSTR2 and analyze the similarities and differences among them. According to the helpful suggestion, we first summarized the published complex structures of SSTR2, and we also included structural perspectives of SSTR2 in response to different types of ligands in the result and discussion sections in the revised manuscript, lines 124-128, 134-143.

Table R1. The up-to-date structures of SSTR2.

PDB ID	Resolution (Å)	Ligand	State	Type of Gα protein	reference
7T10	2.5	SST14	Active	Gi3	Nat Struct Mol Biol
7T11	2.7	Octreotide	Active	Gi3	Nat Struct Mol Biol
7UL5	3.10	—	Inactive	—	Skiniotis To be published
7WIC	2.80	SST14	Active	Gi1	Cell Res
7WIG	2.7	L-054,264	Active	Gi1	Cell Res
7WJ5	3.72	SST14	Active	Gi1	Elife
7XAT	2.85	SST14	Active	Gi1	Cell Discov.
7XAU	2.97	Octreotide	Active	Gi1	Cell Discov.
7XAV	2.87	Lanreotide	Active	Gi1	Cell Discov.
7XMR	3.10	SST14	Active	Gi1	Cell Res

7XN9*	2.60	L-054,522	Intermediate	—	Cell Res
7XNA*	2.65	CYN 154806	Inactive	—	Cell Res
7YAE	3.37	Octreotide	Active	Gi1	This study
7YAC	3.24	Paltusotine	Active	Gi1	This study

* Structures determined by the X-ray diffraction strategy.

Compared with SSTR2-octreotide complex structure at a higher resolution of 2.7 Å reported by *Skiniotis* group (PDB code: 7T11), the SSTR2-octreotide structure determined here displays nearly identical conformation with a RMSD of 0.82 Å for the C α atoms of the receptor (Fig. R8a). In detail, the two octreotide molecules fold the same pose in both structures (Fig. R8b, c), even though SSTR2 couples different Gi proteins (Gi1 or Gi3). Further structural comparison reveals the same recognition manner, in which the key residues (D)-Trp4 and Lys5 of octreotide locate at the bottom of the orthosteric pocket, further triggering extracellular signal transmembrane transduction. The critical microswitches required for receptor activation, including the toggle switch, PIF and DRY motifs, are observed to exhibit the identical conformation upon Gi protein coupling (Fig. R8d). Taken together, the structures of SSTR2-Gi complex bound to octreotide determined by different groups all represented the active signaling complex, there is no significant difference among them except for the extracellular tips of the complex structure due to the dynamic features of the receptor. The available structures basically provide us opportunities to investigate the mechanism of ligand recognition and receptor activation.

Moreover, as the reviewer mentioned, an antagonist-bound inactive structure of SSTR2 has been reported by *Zhao* group (PDB code: 7XNA, *Cell research*), as well as an inactive structure of SSTR2 without ligand has been reported by *Skiniotis* group (PDB code: 7UL5). The antagonist CYN 154806 contains the key residues (D)-Trp8-Lys9-Thr10-Cys11 that is also present in octreotide (in octreotide it is numbered in (D)-Trp4-Lys5-Thr6-Cys7). Subsequently, we compared the inactive state of SSTR2 bound to antagonist CYN 154806 with the active state of SSTR2. Structural comparison reveals that Phe5-(D)-Cys6 of CYN 154806 folds into distinct pose from the corresponding (D)-Phe1-Cys2 in octreotide. More importantly, (D)-Trp4 in octreotide or the hydroxybenzotrile moiety in paltusotine, as the key facets for receptor activation, inserts more deeply than the antagonist CYN 154806, further stabilizing the extracellular regions by packing with TM bundle in the activated SSTR2 structure. In contrast, the equivalent residue (D)-Trp8 in CYN 154806 is tilted and inserts into a different hydrophobic site, thus losing contact with TM6. In addition, CYN 154806 is observed to occupy another extended binding pocket (EBP-2) at the extracellular portion, such interaction of EBP-2 with antagonist might hinder the narrowing of the

extracellular regions of SSTR2 for activation (Fig. R8e).

In all, we have added the comparison of these structures in the discussion section, and analyzed the similarities and divergencies within these structures. These five papers mentioned above have been cited in the revised manuscript.

Fig. R8: Structural comparison of SSTR2-octreotide structure determined in our study with that from *Skiniotis* group and with the inactive structure determined by *Zhao* group.

a: Overall structural comparison of SSTR2-octreotide in our study with SSTR2^{7T11}. SSTR2 in our study is shown as cartoon and colored in green-cyan, octreotide is shown as sticks and colored in light pink; SSTR2 in SSTR2^{7T11} is shown as cartoon and colored in gray, octreotide is shown as sticks and colored in yellow. (see also in Supplementary Fig. 4b)

b: The structure models of octreotide in these two structures.

c: Structural comparison of the residues in the octreotide binding pocket of SSTR2 in these two structures. Key residues in SSTR2 are shown as sticks. (see also in Supplementary Fig. 4c)

d: Conformational comparison of the microswitch residues of SSTR2 in these two structures. Key residues in SSTR2 are shown as sticks.

e: Structural comparison of the paltusotine- (cyan), octreotide- (light pink) and CYN 154806^{7XNA} (orange)-bound structures. Small molecules are shown in sticks and SSTR2 are shown as cartoon. The cycle indicates the extended part of CYN 154806.

The highlight of this work, in my mind, is the studies of β -arrestin recruitment and

internalization. However, I am not totally convinced that N276^{6.55} and F294^{7.35} play key roles to the signal bias of SSTR2 when sensing octreotide. The N276^{6.55}A and F294^{7.35}A mutation indeed cause larger reduction in octreotide stimulated β -arrestin recruitment compared to Gi signaling, but I am not sure if the differences are due to biased signaling or just due to impaired affinity. For example, N276^{6.55}A causes ~90 fold increase in EC50 for octreotide stimulated Gi activation and ~540 fold increase for β -arrestin recruitment. It's possible that this mutation just reduced the affinity of octreotide and made this ligand less functional in activating SSTR2. This affinity reduction has a larger effect on β -arrestin recruitment assay because β -arrestin generally binds weaker to the receptor than G protein.

Response: We thank the reviewer's meaningful question and advice. We totally understand the reviewer's concern. Previous ligand binding assays of SSTR2 were taken in competition with ¹²⁵I-SST14. Unfortunately, we could not get this radioligand due to the long shipping time from abroad during the COVID-19 pandemic.

We applied intramolecular fluorescent arsenical hairpin bioluminescence resonance energy transfer (FIAsH-BRET) method instead to monitor the conformation changes of SSTR2 in response to different types of ligands, especially the extracellular regions occur notable rearrangement upon ligands binding to the orthosteric site²¹, and the results of the measurement can reflect the ligand binding ability with the receptor at a certain degree^{22,23}.

We therefore designed five sites at three extracellular loops (ECL) for incorporating with FIAsH motif (Fig. R9a), and the sensor Nluc was introduced to the N-terminus of SSTR2. The BRET signal between Nluc-N terminus and FIAsH-ECL exhibited a notable increase at position I284^{ECL3} labelling (Fig. R9b). Then, we measured the BRET signals at two different time points after ligand administration. Compared with wild-type SSTR2, the results of 3 min administration from N276^{6.55}A or F294^{7.35}A mutant revealed that the binding of octreotide or paltusotine to the receptor exhibited similar conformation changes, which means that the mutations at position 6.55 and 7.35 did not affect the recognition of octreotide or paltusotine by SSTR2 (Fig. R9c, e). Whereas the results of 9 min administration indicated both mutants decreased the BRET signals when sensing octreotide, by contrast, N276^{6.55}A or F294^{7.35}A mutation of SSTR2 retained similar signals with wild-type SSTR2 in response to paltusotine (Fig. R9d, f). Collectively, N276^{6.55}A or F294^{7.35}A mutation may influence the conformation of SSTR2 in response to different ligands. Consistent with our finding, the results of the competition assays from Zhao group²⁰ reveals that F294^{7.35}A mutation reduced 6-fold binding affinity relative to wild-type SSTR2. In our β -arrestin recruitment assays, we measured the signal after 3 min ligand administration, the results suggested that N276^{6.55}A and F294^{7.35}A mutations were associated with signal bias of SSTR2. Of course, we can't exclude the possibility that these mutations could affect

the binding affinity with the receptor.

Meanwhile, we agree with the reviewer's opinion, the affinity reduction might have effect more on the β -arrestin recruitment assays compared with the G protein signaling. We wondered whether a key residue mutant from SSTR2 could affect the binding affinity and G protein signaling but still retain similar ability to recruit β -arrestin compared to wild-type receptor. It is noteworthy that a mutation Y302^{7.43}A of SSTR2 in the orthosteric site slightly impaired the octreotide induced β -arrestin recruitment, however, the Y302^{7.43}A substitution nearly abolished the binding of octreotide to SSTR2²⁰. Given that the effector β -arrestin coupling to the intracellular portion of SSTR2 is likely to promote the receptor to recognize the extracellular orthosteric ligands in an allosteric manner, mutating the residue that only related to the ligand binding may not affect β -arrestin recruitment in some extent (especially affect β -arrestin recruitment more than G protein signaling). In a word, our results suggest that N276^{6.55} and F294^{7.35} should be related with the β -arrestin signal bias of SSTR2 by octreotide (Fig. R9g-j). We have modified the related description and included the FLA^{SH}-BRET results in the revised manuscript in the section of Result 5 "Signal bias properties of SSTR2 with different ligands" section, lines 286-294.

Fig. R9: FIAsh-BRET assay of SSTR2. (see also in Supplementary Fig. 9)

a: Schematic representation of the FIAsh-BRET assay design. The red mark showed the CCPGCC FIAsh motif was inserted behind the indicated residue. Lower panel: the detailed sequence of the FIAsh motifs.

b: The dose response curve of the five SSTR2 FIAsh-BRET sensors. Data represent mean \pm SEM from three independent experiments.

c-d: The effects of SSTR2 N276^{6.55}A and F294^{7.35}A mutation on FIAsh-BRET assay after 3 min (**c**) or 9 min (**d**) treatment of octreotide. Data represent mean \pm SEM from three independent experiments.

e-f: The effects of SSTR2 N276^{6.55}A and F294^{7.35}A mutation on FIAsh-BRET assay after 3 min (**e**) or 9 min (**f**) treatment of paltusotine. Data represent mean \pm SEM from three independent experiments.

g, h: The effects of SSTR2 N276^{6.55}A, F294^{7.35}A and Y302^{7.43}A mutation on cAMP inhibition (**g**) and β -arrestin recruitment (**h**) induced by octreotide. Data represent mean \pm SEM from three independent experiments.

i, j: The effects of SSTR2 N276^{6.55}A, F294^{7.35}A and Y302^{7.43}A mutation on cAMP inhibition (**i**) and β -arrestin recruitment (**j**) induced by paltusotine. Data represent mean \pm SEM from three independent experiments.

I also noticed that the current β -arrestin recruitment assays were done without co-transfection of GRK. It would be more convincing if the authors could repeat the assay in presence of GRK and check if these two mutations indeed abolish β -arrestin recruitment.

Response: We thank the reviewer's constructive suggestion. Previous study has shown that β -arrestin recruitment by SSTR2 was associated with GRK2 and GRK3, which belonged to the same GRK branch^{24,25}. Therefore, we repeated the β -arrestin recruitment assays with co-transfection of GRK2 according to the suggestion. Consistently, we did not observe significant differences for the efficacies of ligand induced β -arrestin recruitment in the presence or absence of GRK2. For instance, both N276^{6.55}A and F294^{7.35}A mutants of SSTR2 nearly abolished the ability to recruit cellular β -arrestin in response to octreotide, by contrast, these two mutants slightly influenced paltusotine induced β -arrestin recruitment (Fig. R10). We have included the related description in Result 5 "Signal bias properties of SSTR2 with different ligands" section, lines 283-285.

Fig. R10: The influence of GRK in the β -arrestin recruitment.

a, b: The effects of SSTR2 N276^{6.55}A mutation on β -arrestin recruitment with (a) or without (b) co-transfection of GRK2 induced by octreotide.

c, d: The effects of SSTR2 N276^{6.55}A mutation on β -arrestin recruitment with (c) or without (d) co-transfection of GRK2 induced by paltusotine.

e, f: The effects of SSTR2 F294^{7.35}A mutation on β -arrestin recruitment with (e) or without (f) co-transfection of GRK2 induced by octreotide.

g, h: The effects of SSTR2 F294^{7.35}A mutation on β -arrestin recruitment with (g) or without (h) co-transfection of GRK2 induced by paltusotine.

One interesting question is why paltusotine is more G protein biased compared to octreotide, as this information may guide future drug design. But the authors did not

really address this question.

Response: Thanks for the valuable comment. As discussed in our study, the small molecule paltusotine induced lower β -arrestin recruitment relative to octreotide and exerted as a G protein-biased ligand (Supplementary Fig. 8a). In the revised manuscript, we have provided descriptions to address the relationship of signal bias of SSTR2 with future drug design, and further modified the section of “Signal bias properties of SSTR2 with different ligands”.

Briefly, we first compared the two SSTR2 structures in complex with octreotide and paltusotine and carefully analyzed the key residues engaged in ligand recognition. As is shown in Fig. 3b (main figure), we noticed that octreotide occupied an extended binding pocket of SSTR2, subsequently, by generating a range of mutations in the extended binding pocket, we found that alanine substitution of I284^{ECL3}, K291^{7.32}, N276^{6.55} residues resulted in markedly reduced β -arrestin signal (Fig. R11a, c), but only slightly affected the Gi pathway induced by octreotide (Fig. R11b, c). By contrast, the small molecule paltusotine only occupies the core region of orthosteric site in SSTR2, losing the interaction manner that is shown in octreotide binding, displaying lower β -arrestin recruitment and alanine substitution of I284^{ECL3}, K291^{7.32} and N276^{6.55} do not reduce the paltusotine induced β -arrestin recruitment. The contact of the ligand within the extended binding pocket of SSTR2 is likely to be involved in β -arrestin signal bias modulation, thus designing specific drugs target the core region and with minimum contact with the extended binding pocket would facilitate the development of G protein-biased ligand of SSTR2.

Furthermore, we tend to find residues involved in bias regulation in the orthosteric site. Interestingly, our results revealed that replacement of F294^{7.35} with alanine nearly impaired the octreotide induced β -arrestin recruitment (Fig. R11a-c), however, the F294^{7.35}A mutant only slightly influenced the β -arrestin signal of SSTR2 in response to paltusotine (Fig. R11d-f). In addition, structural comparison indicated that the side chain of F294 is closer to octreotide than paltusotine, despite of different rotameric displacement in the two structures. Combined with structural observation, our results of cAMP inhibition as well as β -arrestin recruitment assays enabled us to speculate that F294^{7.35} should be engaged in modulation of β -arrestin recruitment. Collectively, the understanding of the bias properties of SSTR2 would help pharmacologists to design or optimize efficacious small molecules with biased signaling.

Fig. R11: Biased property analyses of the residues within the octreotide or paltusotine binding pocket in SSTR2.

a-c: The effects of SSTR2 I284^{ECL3}, K291^{7.32}, N276^{6.55}A and F294^{7.35} mutations on β -arrestin recruitment(a) and cAMP inhibition (b) induced by octreotide. **c.** Bias factors of these mutants. Statistical differences between wild-type and mutants were determined by one way of variance ANOVA with Dunnett's test. *P < 0.033, ***p < 0.01 n.s., not significant, n.d., not detected. Data represent mean \pm SEM from three independent experiments.

d-f: The effects of SSTR2 I284^{ECL3}, K291^{7.32}, N276^{6.55}A and F294^{7.35} mutations on β -arrestin recruitment(d) and cAMP inhibition (e) induced by paltusotine. **f.** Bias factors of these mutants. Statistical differences between wild-type and mutants were determined by one way of variance ANOVA with Dunnett's test. *P < 0.033, ***p < 0.01 n.s., not significant, n.d., not detected. Data represent mean \pm SEM from three independent experiments.

Other minor suggestions include:

1. In line 29, the authors claim that "drug resistance occurs in a subset of patients, which may be correlated with SSTR subtype selectivity or cell surface expression." I understand the correlation between 'drug resistance' and 'cell surface expression'. But I don't quite understand why 'subtype selectivity' correlates with 'drug resistance'.

Response: We gratefully appreciate the reviewer for pointing out the correlation between drug resistance with subtype selectivity or cell surface expression. We apologize for this ambiguous description about the correlation between them. Subtype selectivity of drug discovery for specific SSTR member is actually valuable therapeutic strategy since five SSTRs are involved in divergent physiological functions, and some progresses have been made in selective SST analogs or small molecules with better

pharmacological profile. We didn't find a clue of the correlation of drug resistance with subtype selectivity from previous literatures, therefore, we revised the sentence in the revised manuscript, line 29.

2. The confocal fluorescence microscopy images are confusing to me. For example, in Fig. 6c and 6d, it looks like paltusotine induces as much internalization as octreotide does in WT SSTR2. While in Fig. 6c, it looks like octreotide induces as much internalization for N276^{6.55}A and F294^{7.35}A mutants compared to WT SSTR2. These results are not consistent with the main conclusion of the manuscript.

Response: We thank the reviewer for pointing out this issue. We checked the raw data carefully and found that we placed the confocal image in the wrong order in the previous version of manuscript and we have corrected it (Fig. R12a, b). We apologize for this confusing figure presentation. In the revised manuscript, to make sure of the authenticity of the data, we repeated the confocal fluorescence microscopy experiments to support our standpoint. As is shown in Fig. R12c and d, in consistent with our previous data, the results revealed that both N276^{6.55}A and F294^{7.35}A mutations showed diminished internalization of SSTR2 when treated with octreotide, while only slightly influenced the internalization under paltusotine treatment.

Fig. R12: Differential internalization activated by octreotide and paltusotine.

a, b: Effects of N276^{6.55}A or F294^{7.35}A mutation on the internalization of SSTR2 in HEK293 cells treated with octreotide (**a**) or paltusotine (**b**) indicated by confocal fluorescence microscopy. eGFP-tagged SSTR2 wild-type or mutant plasmid was transfected in HEK293 cells and then cells were treated with 10 μ M octreotide or paltusotine for 30 min and analyzed by confocal fluorescence microscopy (green, eGFP-SSTR2; blue, DAPI). Scale bar, 50 μ m. (see also in Fig. 6c, d-previous version)

c, d: The newly repeated confocal fluorescence microscopy data. (see also in Fig. 6c, d-revised version)

References:

- 1 Paragliola, R. M., Corsello, S. M. & Salvatori, R. Somatostatin receptor ligands in acromegaly: clinical response and factors predicting resistance. *Pituitary* **20**, 109-115, doi:10.1007/s11102-016-0768-4 (2017).
- 2 Bo, Q. *et al.* Structural insights into the activation of somatostatin receptor 2 by cyclic SST analogues. *Cell Discov* **8**, 47, doi:10.1038/s41421-022-00405-2 (2022).
- 3 Robertson, M. J., Meyerowitz, J. G., Panova, O., Borrelli, K. & Skiniotis, G. Plasticity in ligand recognition at somatostatin receptors. *Nat Struct Mol Biol* **29**, 210-217, doi:10.1038/s41594-022-00727-5 (2022).
- 4 Yang, L. *et al.* Spiro[1H-indene-1,4'-piperidine] derivatives as potent and selective non-peptide human somatostatin receptor subtype 2 (sst2) agonists. *J Med Chem* **41**, 2175-2179, doi:10.1021/jm980194h (1998).
- 5 Chen, L. N. *et al.* Structures of the endogenous peptide- and selective non-peptide agonist-bound SSTR2 signaling complexes. *Cell Res*, doi:10.1038/s41422-022-00669-z (2022).
- 6 Suomivuori, C. M. *et al.* Molecular mechanism of biased signaling in a prototypical G protein-coupled receptor. *Science* **367**, 881-887, doi:10.1126/science.aaz0326 (2020).
- 7 Bouley, R. *et al.* Functional role of the NPxxY motif in internalization of the type 2 vasopressin receptor in LLC-PK1 cells. *Am J Physiol Cell Physiol* **285**, C750-762, doi:10.1152/ajpcell.00477.2002 (2003).
- 8 Xu, Z. *et al.* Structural basis of sphingosine-1-phosphate receptor 1 activation and biased agonism. *Nat Chem Biol* **18**, 281-288, doi:10.1038/s41589-021-00930-3 (2022).
- 9 Namkung, Y. *et al.* Monitoring G protein-coupled receptor and beta-arrestin trafficking in live cells using enhanced bystander BRET. *Nat Commun* **7**, 12178, doi:10.1038/ncomms12178 (2016).
- 10 Smith, J. S. *et al.* C-X-C Motif Chemokine Receptor 3 Splice Variants Differentially Activate Beta-Arrestins to Regulate Downstream Signaling Pathways. *Mol Pharmacol* **92**, 136-150, doi:10.1124/mol.117.108522 (2017).
- 11 Sun, L. & Coy, D. H. Somatostatin and its Analogs. *Curr Drug Targets* **17**, 529-537, doi:10.2174/1389450116666141205163548 (2016).
- 12 Casarini, A. P. *et al.* Acromegaly: correlation between expression of somatostatin receptor subtypes and response to octreotide-lar treatment. *Pituitary* **12**, 297-303, doi:10.1007/s11102-009-0175-1 (2009).
- 13 Gunther, T. *et al.* International Union of Basic and Clinical Pharmacology. CV. Somatostatin

- Receptors: Structure, Function, Ligands, and New Nomenclature. *Pharmacol Rev* **70**, 763-835, doi:10.1124/pr.117.015388 (2018).
- 14 Lesche, S., Lehmann, D., Nagel, F., Schmid, H. A. & Schulz, S. Differential effects of octreotide and pasireotide on somatostatin receptor internalization and trafficking in vitro. *J Clin Endocrinol Metab* **94**, 654-661, doi:10.1210/jc.2008-1919 (2009).
- 15 Fougner, S. L. *et al.* The clinical response to somatostatin analogues in acromegaly correlates to the somatostatin receptor subtype 2a protein expression of the adenoma. *Clin Endocrinol (Oxf)* **68**, 458-465, doi:10.1111/j.1365-2265.2007.03065.x (2008).
- 16 Melmed, S. Acromegaly pathogenesis and treatment. *J Clin Invest* **119**, 3189-3202, doi:10.1172/JCI39375 (2009).
- 17 Gatto, F. *et al.* beta-Arrestin 1 and 2 and G protein-coupled receptor kinase 2 expression in pituitary adenomas: role in the regulation of response to somatostatin analogue treatment in patients with acromegaly. *Endocrinology* **154**, 4715-4725, doi:10.1210/en.2013-1672 (2013).
- 18 Stephen F. Betz, S. M., Ana Karin Kusnetzow, Jon D. Athanacio, Elizabeth Rico-Bautista, Ajay Madan, Michael Johns, Yun Fei Zhu, Agnes Schonbrunn, R. Scott Struthers. Suppression of growth hormone and insulin-like growth factor 1 in rats after oral administration of CRN00808, a small molecule, SST2 selective somatostatin biased agonist. *Endocrine Rev* **39(2)** (2018).
- 19 Heo, Y. *et al.* Cryo-EM structure of the human somatostatin receptor 2 complex with its agonist somatostatin delineates the ligand-binding specificity. *Elife* **11**, doi:10.7554/eLife.76823 (2022).
- 20 Zhao, W. *et al.* Structural insights into ligand recognition and selectivity of somatostatin receptors. *Cell Res* **32**, 761-772, doi:10.1038/s41422-022-00679-x (2022).
- 21 Bokoch, M. P. *et al.* Ligand-specific regulation of the extracellular surface of a G-protein-coupled receptor. *Nature* **463**, 108-112, doi:10.1038/nature08650 (2010).
- 22 Ping, Y. Q. *et al.* Structures of the glucocorticoid-bound adhesion receptor GPR97-Go complex. *Nature* **589**, 620-626, doi:10.1038/s41586-020-03083-w (2021).
- 23 Nuber, S. *et al.* beta-Arrestin biosensors reveal a rapid, receptor-dependent activation/deactivation cycle. *Nature* **531**, 661-664, doi:10.1038/nature17198 (2016).
- 24 Gurevich, V. V. & Gurevich, E. V. GPCR Signaling Regulation: The Role of GRKs and Arrestins. *Front Pharmacol* **10**, 125, doi:10.3389/fphar.2019.00125 (2019).
- 25 Treppiedi, D. *et al.* Somatostatin Receptor Type 2 (SSTR2) Internalization and Intracellular Trafficking in Pituitary GH-Secreting Adenomas: Role of Scaffold Proteins and Implications for Pharmacological Resistance. *Horm Metab Res* **49**, 259-268, doi:10.1055/s-0042-116025 (2017).

Reviewers' Comments:

Reviewer #1:

Remarks to the Author:

The authors have fully addressed my concerns in the revised manuscript. I do not have further questions.

Reviewer #2:

Remarks to the Author:

I don't think the FlaSH-BRET assay is good enough to address the affinity question. The assay only detects the association process of the ligand, while the ligand binding affinity is determined by both association rate and dissociation rate. I don't think the authors need to include the data. After all, measuring the affinity is not the key to address my question. Because the key point of my previous question was not about whether N2766.55 and F2947.35 affect the ligand affinity or not, the key point of my question is if the main conclusion of the manuscript "N2766.55 and F2947.35 in SSTR2 modulate the receptor internalization and desensitization" is correct or not.

It's fine that N2766.55 and F2947.35 cause impaired affinity for octreotide. Anyway, affinity differences could also be part of the reason for the signal bias. But to claim that "N2766.55 and F2947.35 in SSTR2 modulate the receptor internalization and desensitization", one has to prove that "N2766.55 and F2947.35" are different from other residues. For example, the authors may find mutations that affect the EC50 of octreotide by ~100 fold in Gi signaling assay, while do not abolish the β -arrestin signaling. In this way, the author may argue that residues "N2766.55 and F2947.35" are unique and play special roles in β -arrestin recruitment. However, when I looked through supplementary table 2, I don't see many such examples. Instead, I saw some other interesting mutations.

For example, the I130A mutation only reduced the EC50 of octreotide induced Gi signaling by ~3 fold (1.9 nM vs 0.6 nM, Emax 91% vs 100%), but reduced the EC50 of β -arrestin recruitment by ~200 fold (1404 nM vs 7.5 nM, Emax 60% vs 100%). The effect seems to be much more obvious than the N2766.55 mutation. Why did not the authors claim that I130 is the key for receptor internalization and desensitization? Similar argument goes to Q102A mutations, which reduced EC50 for Gi signaling by ~25 fold (15.5 nM vs 0.6 nM, Emax 109% vs 100%), but reduced EC50 for β -arrestin recruitment by 856 fold (6396 nM vs 7.5 nM). For comparison, N2766.55 mutation reduced EC50 for Gi assay by 90 fold and EC50 for β -arrestin assay by 540 fold. It just could not convince me that N2766.55 is a special residue for β -arrestin signaling with the current data.

By the way, I found a lot of mistakes in the supplementary table 2 and 3 through a quick look. I understood as the 'fold of WT' column was calculated by dividing the EC50 of mutant with the EC50 of WT. It seems to be correct for some cases, but make no sense for others. Just to list a few:

Supplementary table 2:

I195A mutation shows a EC50 of 6.5 nM, while the WT shows a EC50 of 0.6 nM. The fold of WT should be ~9. But in the table it says 32.

V103A: 2.85 nM vs 0.6 nM (WT). Fold of WT should be ~4.6, but it says 44.44.

F127A: 0.67 nM vs 0.6 nM. Fold of WT should be 1.1, but it says 13.14.

....

One could easily find such mistakes all over the table. The authors really need to make sure that their data are correct.

Point-to-point response letter

We thank the referees for their valuable time in reviewing our manuscript and the constructive suggestions that they have provided. Please find our responses to the specific comments raised by the reviewers below. We have copied each comment in *Italic*, which is followed by our own point-by-point response in **blue**, including details about the corresponding changes to the manuscript.

Reviewer #1:

The authors have fully addressed my concerns in the revised manuscript. I do not have further questions.

Response: We thank the referee for his/her positive comments for our work.

Reviewer #2:

I don't think the FlaSH-BRET assay is good enough to address the affinity question. The assay only detects the association process of the ligand, while the ligand binding affinity is determined by both association rate and dissociation rate. I don't think the authors need to include the data. After all, measuring the affinity is not the key to address my question. Because the key point of my previous question was not about whether N276^{6.55} and F294^{7.35} affect the ligand affinity or not, the key point of my question is if the main conclusion of the manuscript "N276^{6.55} and F294^{7.35} in SSTR2 modulate the receptor internalization and desensitization" is correct or not.

Response: We appreciated for the referee's comments. We understood the referee's concerns. In the revised manuscript, we carried out in-depth data analysis for mutagenesis studies as well as corresponding functional assays according to the referee's suggestion (detail in the next response). In the last manuscript, we sought to get more information of critical role for these residues, therefore, the FIAsh-BRET method was applied to monitor the conformation changes of SSTR2 in response to different types of ligands, which reflects ligand binding ability at some certain degree. We agree that FIAsh-BRET is not a perfect method to directly address the affinity question, and we removed the results of FIAsh-BRET according to the helpful suggestion.

It's fine that N276^{6.55} and F294^{7.35} cause impaired affinity for octreotide. Anyway, affinity differences could also be part of the reason for the signal bias. But to claim that "N276^{6.55} and F294^{7.35} in SSTR2 modulate the receptor internalization and desensitization", one has to prove that "N276^{6.55} and F294^{7.35}" are different from other

residues. For example, the authors may find mutations that affect the EC50 of octreotide by ~100 fold in Gi signaling assay, while do not abolish the β -arrestin signaling. In this way, the author may argue that residues “N276^{6.55} and F294^{7.35}” are unique and play special roles in β -arrestin recruitment. However, when I looked through supplementary table 2, I don't see many such examples. Instead, I saw some other interesting mutations. For example, the I130A mutation only reduced the EC50 of octreotide induced Gi signaling by ~3 fold (1.9 nM vs 0.6 nM, Emax 91% vs 100%), but reduced the EC50 of β -arrestin recruitment by ~200 fold (1404 nM vs 7.5 nM, Emax 60% vs 100%). The effect seems to be much more obvious than the N276^{6.55} mutation. Why did not the authors claim that I130 is the key for receptor internalization and desensitization? Similar argument goes to Q102A mutations, which reduced EC50 for Gi signaling by ~25 fold (15.5 nM vs 0.6 nM, Emax 109% vs 100%), but reduced EC50 for β -arrestin recruitment by 856 fold (6396 nM vs 7.5 nM). For comparison, N276^{6.55} mutation reduced EC50 for Gi assay by 90 fold and EC50 for β -arrestin assay by 540 fold. It just could not convince me that N276^{6.55} is a special residue for β -arrestin signaling with the current data.

Response: We gratefully appreciate for your valuable comments. As is pointed out by the referee, to claim the residues modulate the internalization and desensitization of receptor, we should prove that such residue influences the effect of β -arrestin recruitment more than Gi signaling activation by comparing with other residues. In this study, our focus is to figure out the residues that contribute to the signal bias of SSTR2 when sensing different types of ligands octreotide and paltusotine, since we found that peptide ligand octreotide behaves stronger β -arrestin recruitment than small molecule ligand paltusotine. To answer this question, our first step focused on the ligand binding pocket in SSTR2, then we tried to find out the mutations prefer to affect the β -arrestin recruitment induced by octreotide but not paltusotine.

As is shown in Fig. R1, by generating a range of mutations in the octreotide and paltusotine binding pocket, we found that alanine substitution of I284^{ECL3}, K291^{7.32}, N276^{6.55} or F294^{7.35} resulted in markedly reduced β -arrestin signal, while slightly affected the Gi protein activation induced by octreotide (Fig. R1a-c); by contrast, these mutants did not reduce or only slightly influenced the β -arrestin signal of SSTR2 in response to paltusotine (Fig. R1d-f). In a word, consistent with our observation, our data indicates that paltusotine loses the interaction with these residues (I284^{ECL3}, K291^{7.32}, N276^{6.55} or F294^{7.35}) engaged in octreotide recognition, displaying reduced β -arrestin recruitment efficacy.

Our result also reveals that another two residues (I130A from PIF motif and Q102^{2.63}A) mentioned by the referee indeed affect the β -arrestin recruitment activated by octreotide (Fig. R1g-i), however they also affected the paltusotine induced β -arrestin signal (Fig. R1j-l), which indicates that these residues should be involved in signal bias

property of SSTR2, and this bias feature is presumably a general feature of the receptor and is not depend on different ligands, we have included related description in the revised manuscript. We did not exclude the essential role of these residues for receptor internalization and desensitization. On the other hand, by carrying out in-depth data analysis, we found that V103^{2.63} forms van der Waals forces with paltusotine, alanine substitution of V103^{2.63} markedly reduced the paltusotine-induced β -arrestin signal while had less impact on Gi signaling, which indicated that V103^{2.63} involves in β -arrestin recruitment under the treatment of paltusotine (Fig. R1g-l). We thank the referee's convenient recommendation and have revised and supplied the discussion about these residues in the revised manuscript (Lines 282-295 and lines 311-315).

Fig. R1: Biased property analyses of the residues within the octreotide or paltusotine binding pocket in SSTR2.

a-c: The effects of SSTR2 I284^{ECL3}, K291^{7.32}, N276^{6.55} and F294^{7.35} mutations on β -arrestin recruitment(**a**) and cAMP inhibition (**b**) induced by octreotide. **c.** Bias factors of these mutants. Statistical differences between wild-type and mutants were determined by one way of variance ANOVA with Dunnett's test. **p <0.02, n.d., not detected. Data represent mean \pm SEM from three independent experiments. β value >0 denotes the mutant affects β -arrestin recruitment more than Gi signaling activation (Due to the great impact on the EC₅₀ value of β -arrestin recruitment, the EC₅₀ of some mutants could not be well-fitted. In this case, **n.d.** could be thought of as the x-axis going infinitely to the right).

d-f: The effects of SSTR2 I284^{ECL3}, K291^{7.32}, N276^{6.55} and F294^{7.35} mutations on β -arrestin recruitment(**d**) and cAMP inhibition (**e**) induced by paltusotine. **f.** Bias factors of these mutants. Statistical differences between wild-type and mutants were determined by one way of variance ANOVA with Dunnett's test. **p <0.02, n.d., not detected. Data represent mean \pm SEM from three independent experiments.

g-i: The effects of SSTR2 I130^{3.40}, Q102^{2.63} and V103^{2.64} mutations on β -arrestin recruitment(**g**) and cAMP inhibition (**h**) induced by octreotide. **i.** Bias factors of these mutants. Statistical differences between wild-type and mutants were determined by one way of variance ANOVA with Dunnett's test. ***p <0.001, n.d., not detected. Data represent mean \pm SEM from three independent experiments.

j-l: The effects of SSTR2 I130^{3.40}, Q102^{2.63} and V103^{2.64} mutations on β -arrestin recruitment(**j**) and cAMP inhibition (**k**) induced by paltusotine. **l.** Bias factors of these mutants. Statistical differences between wild-type and mutants were determined by one way of variance ANOVA with Dunnett's test. n.d., not detected. Data represent mean \pm SEM from three independent experiments.

By the way, I found a lot of mistakes in the supplementary table 2 and 3 through a quick look. I understood as the 'fold of WT' column was calculated by dividing the EC50 of mutant with the EC50 of WT. It seems to be correct for some cases, but make no sense for others. Just to list a few:

Supplementary table 2:

I195A mutation shows a EC50 of 6.5 nM, while the WT shows a EC50 of 0.6 nM. The fold of WT should be ~9. But in the table it says 32.

V103A: 2.85 nM vs 0.6 nM (WT). Fold of WT should be ~4.6, but it says 44.44.

F127A: 0.67 nM vs 0.6 nM. Fold of WT should be 1.1, but it says 13.14.

....

One could easily find such mistakes all over the table. The authors really need to make sure that their data are correct.

Response: We thank the referee so much for taking the time to evaluate the supplementary table. We re-checked the Raw Data for the table and found the reason

mentioned by the referee. Because we did numbers of mutagenesis studies in this study, and it was difficult to complete all the detections at the same time. So, we carried out the experiments in batches. In order to ensure the comparability and reliability of our data, a wild-type group was set up in each batch of measurement, and the “fold of WT” was calculated respectively by dividing the EC₅₀ of wild-type (WT) in each detection. But we had to admit that there are some little differences for different batches setups due to storage time of ligand in freezer incubator, which would lead to some discrepancies (mistakes) for this table. The WT value given in the first column of our table is the most representative data with the largest numbers of repeats, and we did show each WT value in the calculation, but we are definitely sure that the calculation didn’t change our conclusion of this paper.

To make the data result more clearly and reliably, we purchased new ligand and re-tested the mutations in the second-round revision. Fortunately, we overcame the trouble pointed out by the referee, we can obtain repeatable value of WT in each measurement as previously. Therefore, this kind of mistakes in our data was gone in the new table, and the results from new measurements still support our previous standpoints. All assays in this study were performed with triple or more biological replicates.

In all, we apologized for this confusing data presentation. In the revised manuscript, we repeated these experiments and checked through our raw data again and again, make sure everything is correct.

Reviewers' Comments:

Reviewer #2:

Remarks to the Author:

The authors have addressed my concerns.